# FORMALIZING GENERALIZATION AND ROBUSTNESS OF NEURAL NETWORKS TO WEIGHT PERTURBATIONS

## ABSTRACT

Studying the sensitivity of weight perturbation in neural networks and its impacts on model performance, including generalization and robustness, is an active research topic due to its implications on a wide range of machine learning tasks such as model compression, generalization gap assessment, and adversarial attacks. In this paper, we provide the first formal analysis for feed-forward neural networks with non-negative monotone activation functions against norm-bounded weight perturbations, in terms of the robustness in pairwise class margin functions and the Rademacher complexity for generalization. We further design a new theory-driven loss function for training generalizable and robust neural networks against weight perturbations. Empirical experiments are conducted to validate our theoretical analysis. Our results offer fundamental insights for characterizing the generalization and robustness of neural networks against weight perturbations.

## 1 INTRODUCTION

Neural network is currently the state-of-the-art machine learning model in a variety of tasks, including computer vision, natural language processing, and game-playing, to name a few. In particular, feed-forward neural networks consists of layers of trainable model weights and activation functions with the premise of learning informative data representations and the complex mapping between data samples and the associated labels. Albeit attaining superior performance, the need for studying the sensitivity of neural networks to weight perturbations is also intensifying owing to several practical motivations. For instance, in model compression, the robustness to weight quantification is crucial for reducing memory storage while retaining model performance (Hubara et al., 2017; Weng et al., 2020). The notion of weight perturbation sensitivity is also used as a metric to evaluate the generalization gap at local minima (Keskar et al., 2017; Neyshabur et al., 2017). In adversarial robustness and security, weight sensitivity can be leveraged as a vulnerability for fault injection and causing erroneous prediction (Liu et al., 2017; Zhao et al., 2019). However, while weight sensitivity plays an important role in many machine learning tasks and problem setups, theoretical characterization of its impacts on generalization and robustness of neural networks remains elusive.

This paper bridges this gap by developing a novel theoretical framework for understanding the generalization gap (through Rademacher complexity) and the robustness (through classification margin) of neural networks against norm-bounded weight perturbations. Specifically, we consider the multiclass classification problem setup and multi-layer feed-forward neural networks with non-negative monotonic activation functions. Our analysis offers fundamental insights into how weight perturbation affects the generalization gap and the pairwise class margin. To the best of our knowledge, this study is the first work that provides a comprehensive theoretical characterization of the interplay between weight perturbation, robustness in classification margin, and generalization gap. Moreover, based on our analysis, we propose a theory-driven loss function for training generalizable and robust neural networks against norm-bounded weight perturbations. We validate its effectiveness via empirical experiments. We summarize our main contributions as follows.

- We study the robustness (worst-case bound) of the pairwise class margin function against weight perturbations in neural networks, including the analysis of single-layer (Theorem 1), all-layer (Theorem 2), and selected-layer (Theorem 3) weight perturbations.
- We characterize the generalization behavior of robust surrogate loss for neural networks under weight perturbations (Section 3.4) through Rademacher complexity (Theorem 4).

- We propose a theory-driven loss design for training generalizable and robust neural networks (Section 3.5). The empirical results in Section 4 validate our theoretical analysis and demonstrate the effectiveness of improving generalization and robustness against weight perturbations.

## 2 RELATED WORKS

In model compression, the robustness to weight quantization is critical to reducing memory size and accesses for low-precision inference and training (Hubara et al., 2017). Weng et al. (2020) showed that incorporating weight perturbation sensitivity into training can better retain model performance (standard accuracy) after quantization. For studying the generalization of neural networks, Keskar et al. (2017) proposed a metric called sharpness (or weight sensitivity) by perturbing the learned model weights around the local minima of the loss landscape for generalization assessment while An (1996) introduced weight noise into the training process and concluded that random noise training improves the overall generalization. Neyshabur et al. (2017) made a connection between sharpness and PAC-Bayes theory and found that some combination of sharpness and norms on the model weights may capture the generalization behavior of neural networks. Additionally, Bartlett et al. (2017) discovered normalized margin measure to be useful towards quantifying generalization property and a bound was therefore constructed to give an quantitative description on the generalization gap. Moreover, Golowich et al. (2019) incorporated additional assumptions to offer tighter and size-independent bounds from the setting of (Neyshabur et al., 2015) and (Bartlett et al., 2017) respectively. Despite development of various generalization bounds, empirical observations in (Nagarajan & Kolter, 2019) showed that once the size of the training dataset grows, generalization bounds proposed in (Neyshabur et al., 2017) and (Bartlett et al., 2017) will enlarge thus become vacuous. Discussions on the relation between (Nagarajan & Kolter, 2019) and our works can be found at Appendix E, where we show that in our studied setting the associated generalization bounds are non-vacuous. On the other hand, Barron & Klusowski (2018) and Theisen et al. (2019) applied several techniques in tandem with a probabilistic method named path sampling to construct a representing set of given neural networks for approximating and studying the generalization property. Another approach considered by Petzka et al. (2020) consists of segmenting the neural networks into two functions, predictor and feature selection respectively where two measures (representativeness and feature robustness) concerning these aforementioned functions were later combined to offer a meaningful generalization bound. However, these works only focused on the generalization behavior of the local minima and did not consider the generalization and robustness under weight perturbations. Weng et al. (2020) proposed a certification method for weight perturbation retaining consistent model prediction. While the certification bound can be used to train robust models with interval bound propagation (Gowal et al., 2019), it requires additional optimization subroutine and computation costs when comparing to our approach. Moreover, the convoluted nature of certification bound complicates the analysis when studying generalization, which is one of our main objectives. In adversarial robustness, fault-injection attacks are known to inject errors to model weights at the inference phase and causing erroneous model prediction (Liu et al., 2017; Zhao et al., 2019), which can be realized at the hardware level by changing or flipping the logic values of the corresponding bits and thus modifying the model parameters saved in memory (Barenghi et al., 2012; Van Der Veen et al., 2016). Zhao et al. (2020) proposed to use the mode connectivity of the model parameters in the loss landscape for mitigating such weight-perturbation-based adversarial attacks.

Although, to the best of our knowledge, theoretical characterization of generalization and robustness for neural networks against *weight* perturbations remains elusive, recent works have studied these properties under another scenario – the *input* perturbations. Both empirical and theoretical evidence have been given to the existence of a fundamental trade-off between generalization and robustness against norm-bounded input perturbations (Xu & Mannor, 2012; Su et al., 2018; Zhang et al., 2019; Tsipras et al., 2019). The adversarial training proposed in (Madry et al., 2018) is a popular training strategy for training robust models against input perturbations, where a min-max optimization principle is used to minimize the worst-case input perturbations of a data batch during model parameter updates. For adversarial training with input perturbations, Wang et al. (2019) proved its convergence and Yin et al. (2019) derived bounds on its Rademacher complexity for generalization. Different from the case of input perturbation, we note that min-max optimization on neural network training subject to weight perturbation is not straightforward, as the minimization and maximization steps are both taken on the model parameters. In this paper, we disentangle the min-max formulation for

weight perturbation by developing bounds for the inner maximization step and provide quantifiable metrics for training generalization and robust neural networks against weight perturbations.

## 3 MAIN RESULTS

We provide an overview of the presentation flow for our main results as follows. First, we introduce the mathematical notations and preliminary information in Section 3.1. In Section 3.2, we establish our weight perturbation analysis on a simplified case of single-layer perturbation. We then use the single-layer analysis as a building block and extend the results to the multi-layer perturbation setting in Section 3.3. In Section 3.4, we define the framework of robust training with surrogate loss and study the generalization property using Rademacher complexity. Finally, we propose a theory-driven loss toward training robust and generalizable neural networks in Section 3.5.

### 3.1 NOTATION AND PRELIMINARIES

**Notation** We start by introducing the mathematical notations used in this paper. We define the set $[L] := \{1, 2, ..., L\}$. For any two non-empty sets $A$, $B$, $\mathbb{F}_{A \mapsto B}$ denotes the set of all functions from $A$ to $B$. We mark the indicator function of an event E as $\mathbb{1}(E)$, which is 1 if $E$ holds and 0 otherwise. We use $sgn(\cdot)$ to denote element-wise sign function that outputs 1 when input is nonnegative and -1 otherwise. Boldface lowercase letters are used to denote vectors (e.g., $\mathbf{x}$), and the $i$-th element is denoted as $[\mathbf{x}]_i$. Matrices are presented as boldface uppercase letters, say $\mathbf{W}$. Given a matrix $\mathbf{W} \in \mathbb{R}^{k \times d}$, we write its $i$-th row, $j$-th column and $(i, j)$ element as $W_{i,:}$, $W_{:,j}$, and $W_{i,j}$ respectively. Moreover, we write its transpose matrix as $(\mathbf{W})^T$. The matrix $(p, q)$ norm is defined as $\|\mathbf{W}\|_{p,q} := \left\| \left[ \|W_{:,1}\|_p, \|W_{:,2}\|_p, ..., \|W_{:,d}\|_p \right] \right\|_q$ for any $p, q \geq 1$. For convenience, we have $\|\mathbf{W}\|_p = \|\mathbf{W}\|_{p,p}$ and write the spectral norm and Frobenius norm as $\|\mathbf{W}\|_\sigma$ and $\|\mathbf{W}\|_F$ respectively. We mark one matrix norm commonly used in this paper – the matrix $(1, \infty)$ norm. With a matrix $\mathbf{W}$, we express its matrix $(1, \infty)$ norm as $\|\mathbf{W}\|_{1,\infty}$, which is defined as $\|\mathbf{W}\|_{1,\infty} = \max_j \|W_{:,j}\|_1$ and $\|\mathbf{W}^T\|_{1,\infty} = \max_i \|W_{i,:}\|_1$. We use $\mathbb{B}_{\mathbf{W}}^\infty(\epsilon)$ to express an element-wise $\ell_\infty$ norm ball of matrix $\mathbf{W}$ within radius $\epsilon$, i.e., $\mathbb{B}_{\mathbf{W}}^\infty(\epsilon) = \{\hat{\mathbf{W}} \mid |\hat{W}_{i,j} - W_{i,j}| \leq \epsilon, \forall i \in [k], j \in [d]\}$.

**Preliminaries** In order to formally explain our theoretical results, we introduce the considered learning problem, neural network model and complexity definition. Let $\mathcal{X}$ and $\mathcal{Y}$ be the feature space and label space, respectively. We place the assumption that all data are drawn from an unknown distribution $\mathcal{D}$ over $\mathcal{X} \times \mathcal{Y}$ and each data point is generated under i.i.d condition. In this paper, we specifically consider the feature space $\mathcal{X}$ as a subset of $d$-dimensional Euclidean space, i.e., $\mathcal{X} \subseteq \mathbb{R}^d$. We denote the symbol $\mathcal{F} \subseteq \mathbb{F}_{\mathcal{X} \mapsto \mathcal{Y}}$ to be the hypothesis class which we use to make predictions. Furthermore, we consider a loss function $\ell : \mathcal{X} \times \mathcal{Y} \longrightarrow [0, 1]$ and compose it with the hypothesis class to make a function family written as $\ell_\mathcal{F} := \{(\mathbf{x}, y) \longrightarrow \ell(f(\mathbf{x}), y) \mid f \in \mathcal{F}\}$. The optimal solution of this learning problem is a function $f^* \in \mathcal{F}$ such that it minimizes the population risk $R(f) = E_{(\mathbf{x},y) \sim \mathcal{D}}[\ell(f(\mathbf{x}), y)]$. However, since the underlying data distribution is generally unknown, one typically aims at reducing the empirical risk evaluated by a set of training data $\{(\mathbf{x}_i, y_i)\}_{i=1}^n$, which can be expressed as $R_n(f) = \frac{1}{n} \sum_{i=1}^n \ell(f(\mathbf{x}_i), y_i)$. The generalization error is the gap between population and empirical risk, which could serve as an indicator of model's performance under unseen data from identical distribution $\mathcal{D}$.

To study the generalization error, one would explore the learning capacity of a certain hypothesis class. In this paper, we adopt the notion of Rademacher complexity as a measure of learning capacity, which is widely used in statistical machine learning literature (Mohri et al., 2018). The empirical Rademacher complexity of a function class $\mathcal{F}$ given a set of samples $\mathcal{S} = \{(\mathbf{x}_i, y_i)\}_{i=1}^n$ is

$$\mathcal{R}_\mathcal{S}(\ell_\mathcal{F}) = E_\nu[\sup_{f \in \mathcal{F}} \frac{1}{n} \sum_{i=1}^n \nu_i \ell(f(\mathbf{x}_i), y_i)] \tag{1}$$

where $\{\nu_i\}_{i=1}^n$ is a set of i.i.d Rademacher random variables with $\mathbb{P}\{\nu_i = -1\} = \mathbb{P}\{\nu_i = +1\} = \frac{1}{2}$. The empirical Rademacher complexity measures on average how well a function class $\mathcal{F}$ correlates with random noises on dataset $\mathcal{S}$. Thus, a richer or more complex family could better correlate with random noise on average. With Rademacher complexity as a toolkit, one can develop the following relationship between generalization error and complexity measure. Specifically, it is shown in

(Mohri et al., 2018) that given a set of training samples $\mathcal{S}$ and assume that the range of loss function $\ell(f(\mathbf{x}), y)$ is $[0, 1]$. Then for any $\delta \in (0, 1)$, with at least probability $1 - \delta$ we have $\forall f \in \mathcal{F}$

$$R(f) \leq R_n(f) + 2\mathcal{R}_\mathcal{S}(\ell_\mathcal{F}) + 3\sqrt{\frac{\log\frac{2}{\delta}}{2n}} \tag{2}$$

Note that when the Rademacher complexity is small, it is then viable to learn the hypothesis class $\mathcal{F}$ by minimizing the empirical risk and thus effectively reducing the generalization gap.

Finally, we define the structure of neural networks and introduce a few related quantities. The problem studied in this paper is a multi-class classification task with the number of classes being $K$. Consider an input vector $\mathbf{x} \in \mathcal{X} \subseteq \mathbb{R}^d$, an L-layer neural network is defined as

$$f_{\mathbf{W}}(\mathbf{x}) = \mathbf{W}^L(...\rho(\mathbf{W}^1\mathbf{x})...) \in \mathbb{F}_{\mathcal{X} \mapsto \mathbb{R}} \tag{3}$$

with $\mathbf{W}$ being the set containing all weight matrices, i.e., $\mathbf{W} := \{\mathbf{W}^i | \forall i \in [L]\}$, and the notation $\rho(\cdot)$ is used to express any non-negative monotone activation function and we further assume that $\rho(\cdot)$ is 1-Lipschitz, which includes popular activation functions such as ReLU applied element-wise on a vector. Moreover, the $i$-th component of neural networks' output is written as $f_{\mathbf{W}}^i(\mathbf{x}) = [f_{\mathbf{W}}(\mathbf{x})]_i$ and a pairwise margin between $i$-th and $j$-th class, denoted as $f_{\mathbf{W}}^{ij}(\mathbf{x}) := f_{\mathbf{W}}^j(\mathbf{x}) - f_{\mathbf{W}}^j(\mathbf{x})$, is said to be the difference between two classes in output of the neural network. Lastly, we use the notion of $\mathbf{z}^k$ and $\hat{\mathbf{z}}^k$ to represent the output vector of the $k$-th layer ($k \in [L-1]$) under natural and weight perturbed settings respectively, which are $\mathbf{z}^k = \rho(\mathbf{W}^k(...\rho(\mathbf{W}^1\mathbf{x})...))$ and $\hat{\mathbf{z}}^k = \rho(\hat{\mathbf{W}}^k(...\rho(\hat{\mathbf{W}}^1\mathbf{x})...))$, where $\hat{\mathbf{W}}^i \in \mathbb{B}_{\mathbf{W}^i}^\infty(\epsilon_i)$ denotes the perturbed weight matrix bounded by its element-wise $\ell_\infty$-norm with radius $\epsilon_i$ for some $i \in [k]$.

## 3.2 BUILDING BLOCK: SINGLE-LAYER WEIGHT PERTURBATION

We study the sensitivity of neural network to weight perturbations through the pairwise margin bound $f_{\mathbf{W}}^{ij}(\mathbf{x})$. Specifically, when $i$ and $j$ corresponds to the top-1 and the second-top class prediction of $\mathbf{x}$, respectively, the margin can be used as an indicator of robust prediction under weight perturbation to $\mathbf{W}$. For ease of understanding, we first consider a simple example with a three-layer neural network and explain the bound through the error propagation incurred by weight perturbation.

We define the neural network as $f_{\mathbf{W}}(\mathbf{x}) = \mathbf{W}^3\rho(\mathbf{W}^2\rho(\hat{\mathbf{W}}^1\mathbf{x}))$ with $\mathbf{W}^i$ being the weight matrix of the $i$-th layer and assume that one could only perturb any element in the first weight matrix within an $\ell_\infty$ norm ball of radius $\epsilon$, i.e., $\hat{\mathbf{W}}^1 \in \mathbb{B}_{\mathbf{W}^1}^\infty(\epsilon)$. We also define an error vector as $\mathbf{e}_i$, which stands for the entry-wise error after propagating through the $i$-th layer. Since no perturbations happened prior to the first layer, we would directly take input vector $\mathbf{x}$ and derive an upper bound on the entry-wise error $\mathbf{e}_1$. While every element in the first weight matrix is allowed to change its magnitude by at most $\epsilon$, the maximum error for any entry by matrix-vector multiplication becomes

$$[\mathbf{e}_1]_i := |\hat{W}_{i,:}^1\mathbf{x} - W_{i,:}\mathbf{x}| \leq \sum_j |\hat{W}_{i,j}^1 - W_{i,j}^1||[\mathbf{x}]_j| \leq \sum_j \epsilon|[\mathbf{x}]_j| = \epsilon\|\mathbf{x}\|_1. \tag{4}$$

Since the following layer weight is not subject to perturbation, we simply take the magnitude of each element in the subsequent weight matrix to calculate the next error vector. In this case, we have the next layer's error $\mathbf{e}_2$ with $\mathbf{W}^2$ as $[\mathbf{e}_2]_i = \sum_j |W_{i,j}^2|[\mathbf{e}_1]_j = \epsilon\|\mathbf{x}\|_1 \sum_j |W_{i,j}^2|$. Eventually, with error propagation over layers, we arrive at the final layer and are able to assess the maximum change of any entry in output value. By recalling the pairwise class margin $f_{\mathbf{W}}^{ij}(\mathbf{x})$, we would like to inspect the relative change in error between any two classes. Specifically, we derive an upper bound on the pairwise margin between any two classes $\alpha$ and $\beta$. In the above example, the difference in entry-wise maximum error can be deduced in the following manner:

$$[\mathbf{e}_3]_\alpha - [\mathbf{e}_3]_\beta = \sum_k (|W_{\alpha,k}^3| - |W_{\beta,k}^3|)[\mathbf{e}_2]_k \overset{(i)}{\leq} \sum_k (|W_{\alpha,k}^3 - W_{\beta,k}^3|)\epsilon\|\mathbf{x}\|_1 \sum_l |W_{k,l}^2| \tag{5}$$

$$\overset{(ii)}{\leq} \epsilon\|\mathbf{x}\|_1 \, max_k \|W_{k,:}^2\|_1 \sum_k (|W_{\alpha,k}^3 - W_{\beta,k}^3|) = \epsilon\|\mathbf{x}\|_1 \left\|(\mathbf{W}^2)^T\right\|_{1,\infty} \left\|W_{\alpha,:}^3 - W_{\beta,:}^3\right\|_1, \tag{6}$$

where inequality $(i)$ comes from triangle inequality and inequality $(ii)$ results from taking the row in $\mathbf{W}^2$ with maximum $\ell_1$ norm. It is worth noting that there exist possible scenarios for the above inequalities to hold and therefore achieving the worst-case error. Specifically, using the example in Section 3.2, as we trace down the associated inequality bound in (i), we see that the first inequality can be achieved when the final weight layer possesses all positive weights and that the row associated with label $\alpha$ is greater than label $\beta$ in all individual entries. Furthermore, as long as the second weight matrix $\mathbf{W}^2$ has equal $\ell_1$ norm throughout all rows, we can then tighten the bound to give the worst-case error in (ii). Nevertheless, we could see from the above example that the difference of maximum error between entries in output would be propagating at the rate of weight matrices' $(1, \infty)$ norm. By utilizing this essential concept, we introduce the first theorem of our results, which provides an upper bound on the pairwise margin under single-layer weight perturbation.

**Theorem 1 ($N$-th layer weight perturbation ($N \neq L$))** *Let $f_{\mathbf{W}}(\mathbf{x}) = \mathbf{W}^L(...\rho(\mathbf{W}^1\mathbf{x})...)$ denote an $L$-layer neural network and let $f_{\widehat{\mathbf{W}}}(\mathbf{x}) = \mathbf{W}^L(..\hat{\mathbf{W}}^N...\rho(\mathbf{W}^1\mathbf{x})...)$ with $\hat{\mathbf{W}}^N \in \mathbb{B}^\infty_{\mathbf{W}^N}(\epsilon), N \neq L$, denote the corresponding network subject to $N$-th layer perturbation. For any set of perturbed and unperturbed pairwise margin $f^{ij}_{\widehat{\mathbf{W}}}(\mathbf{x})$ and $f^{ij}_{\mathbf{W}}(\mathbf{x})$, we have*

$$f^{ij}_{\widehat{\mathbf{W}}}(\mathbf{x}) \leq f^{ij}_{\mathbf{W}}(\mathbf{x}) + \epsilon \left\|W^L_{i,:} - W^L_{j,:}\right\|_1 \left\|\mathbf{z}^{N-1}\right\|_1 \Pi^{L-N-1}_{k=1} \left\|(\mathbf{W}^{L-k})^T\right\|_{1,\infty}$$

*where $\mathbf{z}^k = \rho(\mathbf{W}^k(...\rho(\mathbf{W}^1\mathbf{x})...))$.*

*Proof*: See Appendix A.1

Since the final layer does not have any activation function, the margin bound on the margin difference when only perturbing the final layer can be simply derived, which is given in the following lemma.

**Lemma 1 (Final-layer weight perturbation)** *Consider the case $N = L$ in Theorem 1, we have $f^{ij}_{\widehat{\mathbf{W}}}(\mathbf{x}) \leq f^{ij}_{\mathbf{W}}(\mathbf{x}) + 2\epsilon \left\|\mathbf{z}^{L-1}\right\|_1$, where $\mathbf{z}^{L-1}$ is the output of the $(L-1)$-th layer.*

*Proof*: See Appendix A.1.

### 3.3 GENERAL SETTING: MULTI-LAYER WEIGHT PERTURBATION

With the developed single-layer analysis in Section 3.2 as a building block, we now extend our analysis to the general setting of multi-layer weight perturbation, which is further divided into two cases: (i) the case of perturbing all $L$ layers; and (ii) the case of perturbing $I$ out of $L$ layers.

#### 3.3.1 PERTURBING ALL LAYERS

Once equipped with the concept of error propagation over subsequent layers, we consider the scenario where every layer in a neural network is subject to weight perturbation. We denote the model under this circumstance as the all-perturbed setting. The following theorem states an upper bound on the pairwise margin between the natural (unperturbed) and all-perturbed settings.

**Theorem 2 (all-layer perturbation)** *Let $f_{\mathbf{W}}(\mathbf{x}) = \mathbf{W}^L(...\rho(\mathbf{W}^1\mathbf{x})...)$ denote an $L$-layer (natural) neural network and let $f_{\widehat{\mathbf{W}}}(\mathbf{x}) = \hat{\mathbf{W}}^L(..\hat{\mathbf{W}}^N...\rho(\hat{\mathbf{W}}^1\mathbf{x})...)$ with $\hat{\mathbf{W}}^m \in \mathbb{B}^\infty_{\mathbf{W}^m}(\epsilon_i), \forall m \in [L]$, denote its perturbed version. For any set of pairwise margin $f^{ij}_{\widehat{\mathbf{W}}}(\mathbf{x})$ and $f^{ij}_{\mathbf{W}}(\mathbf{x})$, we have*

$$f^{ij}_{\widehat{\mathbf{W}}}(\mathbf{x}) \leq f^{ij}_{\mathbf{W}}(\mathbf{x}) + \left\|W^L_{i,:} - W^L_{j,:}\right\|_1 \left\{ \underbrace{\epsilon_1 \|\mathbf{x}\|_1 \Pi^{L-2}_{l=1} \left\|(\mathbf{W}^{L-l})^T\right\|_{1,\infty}}_{\text{Input Layer Error}} + \right.$$

$$\left. \underbrace{\sum^{L-3}_{k=1} \left(\Pi^{L-1}_{m=k+2} \left\|(\mathbf{W}^m)^T\right\|_{1,\infty}\right) \epsilon_{k+1} \left\|\mathbf{z}^{k*}\right\|_1}_{\text{Error from layer 1 to L-2}} + \underbrace{\epsilon_{L-1} \left\|\mathbf{z}^{L-2*}\right\|_1}_{\text{Error of layer L-1}} \right\} + \underbrace{2\epsilon_L \left\|\mathbf{z}^{L-1*}\right\|_1}_{\text{Error of layer L}}$$

*where $\mathbf{z}^{k*} = \rho(\mathbf{W}^{k*}...\rho(\mathbf{W}^{1*}\mathbf{x})$ with $\mathbf{W}^{m*}$ defined as* $\begin{cases} W^{m*}_{i,j} = W^m_{i,j} + \epsilon_m, \forall i, j \text{ and } \forall m \in [L] \setminus \{1\} \\ W^{1*}_{i,j} = W^1_{i,j} + sgn([\mathbf{x}]_j) \epsilon_1, \forall i, j \end{cases}$

*Proof*: See Appendix A.2.2.

Here, we provide some intuition on deriving the upper bound of the margin in the all-perturbed setting. The scheme behind this all-perturbed scenario can be viewed as an inductive layer-wise error propagation. Specifically, we can choose any perturbed layer as the commencement point of propagation, then fix any other weight matrices' values and further calculate the propagation of error from that layer using the concept in Section 3.2. In such manner, after iterating through all these weight matrices subject to weight perturbation, one could obtain the final change in output value and therefore establish the pairwise margin bound. A close inspection of the bound shows that the propagation of error causes the first term since the input layer and the rest of the terms are errors propagating since the $i$-th layer in the neural network, where $i \in [L]$.

### 3.3.2 Perturbing Multiple Layers

The all-perturbed setting is a special case of perturbing layers from an index set $I$ when $I = [L]$. We extend our analysis to the general multi-layer weight perturbation setting with $I \subseteq [L]$, which includes the single-layer setting ($I = \{N\}$) and all-perturbed setting ($I = [L]$) as special cases.

**Theorem 3 (multiple-layer perturbation)** *Let an L-Layer neural network be written as $f_{\boldsymbol{W}}(\boldsymbol{x}) = \boldsymbol{W}^L(...\rho(\boldsymbol{W}^1\boldsymbol{x})...)$. Given an index set $I \subseteq [L]$, we define the perturbed neural network as*

$$f_{\widehat{\boldsymbol{W}}}(\boldsymbol{x}) = \tilde{\boldsymbol{W}}^L(...\tilde{\boldsymbol{W}}^N...\rho(\tilde{\boldsymbol{W}}^1\boldsymbol{x})...) \; with \begin{cases} \tilde{\boldsymbol{W}}^i = \boldsymbol{W}^i, \; \forall i \in [L] \setminus I \\ \tilde{\boldsymbol{W}}^i = \hat{\boldsymbol{W}}^i, \; \hat{\boldsymbol{W}}^i \in \mathbb{B}_{\boldsymbol{W}^i}^\infty(\epsilon_i), \forall i \in I \end{cases}$$

*for any pairwise margin between $f_{\widehat{\boldsymbol{W}}}^{ij}(\boldsymbol{x})$ and $f_{\boldsymbol{W}}^{ij}(\boldsymbol{x})$ we have*

$$f_{\widehat{\boldsymbol{W}}}^{ij}(\boldsymbol{x}) \le f_{\boldsymbol{W}}^{ij}(\boldsymbol{x}) + \left\| W_{i,:}^L - W_{j,:}^L \right\|_1 \left\{ \sum_{\ell \in I \setminus \{L, L-1\}} \epsilon_\ell \left\| \boldsymbol{z}^{\ell-1^*} \right\|_1 \left( \Pi_{j=\ell+1}^{L-1} \left\| (\boldsymbol{W}^j)^T \right\|_{1,\infty} \right) + \mathbb{1}(L-1 \in$$
$$I)\epsilon_{L-1} \left\| \boldsymbol{z}^{L-2^*} \right\|_1 \right\} + \mathbb{1}(L \in I) 2\epsilon_L \left\| \boldsymbol{z}^{L-1^*} \right\|_1 := f_{\boldsymbol{W}}^{ij}(\boldsymbol{x}) + \eta_{\boldsymbol{W}}^{ij}(\boldsymbol{x}|I)$$

$$where \; \boldsymbol{z}^{k^*} = \rho(\boldsymbol{W}^{k^*}...\rho(\boldsymbol{W}^{1^*}\boldsymbol{x}))$$

$$with \; \boldsymbol{W}^{m^*} \; defined \; as \begin{cases} W_{i,j}^{m^*} = \begin{cases} W_{i,j}^m + \epsilon_m, \; \forall i,j \; \forall m \in [L] \cap I \setminus \{1\} \\ W_{i,j}^m, \; \forall i,j \; \forall m \in [L] \setminus (I \cup \{1\}) \end{cases} \\ W_{i,j}^{1^*} = \begin{cases} W_{i,j}^1 + sgn([\boldsymbol{x}]_j)\epsilon_1, \; \forall i,j \; if \; 1 \in I \\ W_{i,j}^1, \; \forall i,j \quad otherwise \end{cases} \end{cases} \quad and \; \boldsymbol{z}^{0^*} = \boldsymbol{x}$$

*Proof*: See Appendix A.2.3.

### 3.4 Surrogate Loss and Generalization Bound

#### 3.4.1 Construction of Robust Surrogate Loss on Pairwise Margin

We aim to construct a surrogate loss function based on a standard loss function and study its behavior against weight perturbations. Specifically, given a perturbation radius $\epsilon$ and the original loss function $\ell(f_{\mathbf{W}}(\mathbf{x}), y)$, robust training aims to minimize the following objective function:

$$\tilde{\ell}(f_{\boldsymbol{W}}(\mathbf{x}), y) = max_{\forall m \in [L], \hat{\mathbf{W}}^m \in \mathbb{B}_{\hat{\mathbf{W}}^m}^\infty(\epsilon)} \ell(f_{\widehat{\boldsymbol{W}}}(\mathbf{x}), y) \tag{7}$$

which we call it as the robustness (worst-case) loss. Even for a single data point $(\mathbf{x}, y)$, it is hard to assess the exact robustness loss since it requires the maximization of a non-concave function over a norm ball. To make the problem of robust training against weight perturbations more computationally tractable, we aim to design a surrogate loss as an upper bound on the worst-case loss.

We focus on the construction of surrogate loss by means of pairwise margin bound in Section 3.3. We first define mathematically two popular loss functions in the classification problem, ramp loss and cross entropy, and derive their surrogate versions. Define the margin function $M(f_{\mathbf{W}}(\mathbf{x}), y)$ as

$$M(f_{\boldsymbol{W}}(\mathbf{x}), y) = \min_{y' \ne y} [f_{\boldsymbol{W}}(\mathbf{x})]_y - [f_{\boldsymbol{W}}(\mathbf{x})]_{y'} = [f_{\boldsymbol{W}}(\mathbf{x})]_y - \max_{y' \ne y} [f_{\boldsymbol{W}}(\mathbf{x})]_{y'} \tag{8}$$

The ramp loss for a given data point $(\mathbf{x}, y)$ and neural network $f_{\boldsymbol{W}}(\cdot)$ is written as $\ell_{\text{ramp}}(f_{\boldsymbol{W}}(\mathbf{x}), y) = \phi_\gamma(M(f_{\boldsymbol{W}}(\mathbf{x}), y))$, where the function $\phi_\gamma : \mathbb{R} \mapsto [0, 1]$ is defined as $\phi_\gamma(t) = 1$ if $t \leq 0$, $\phi_\gamma(t) = 0$ if $t \geq \gamma$, and $\phi_\gamma(t) = 1 - \frac{t}{\gamma}$ if $t \in [0, \gamma]$. Since the ramp loss is a piece-wise linear function, its surrogate loss can be directly obtained with the pairwise margin bound in Section 3.3. The cross entropy is written as $CE(\tilde{f}_{\boldsymbol{W}}(\mathbf{x}), y) = -\ln([\tilde{f}_{\boldsymbol{W}}(\mathbf{x})]_y)$, where $\tilde{f}_{\boldsymbol{W}}(\mathbf{x})$ represents a neural network with its output passing through a softmax layer. That is, $[\tilde{f}_{\boldsymbol{W}}(\mathbf{x})]_i = \frac{\exp[f_{\boldsymbol{W}}(\mathbf{x})]_i}{\sum_{k \in [K]} \exp[f_{\boldsymbol{W}}(\mathbf{x})]_k}$.

For ease of demonstration, we will be using ramp loss and its pairwise margin under single-layer perturbation in the following lemma. The surrogate loss analysis for cross entropy and robust surrogate loss for multiple-layer perturbation is given in Appendix B.

**Lemma 2 (robust surrogate ramp loss)** *Let $N \in [L]$ denote the perturbed layer index and let*

$$\hat{\ell}(f_{\boldsymbol{W}}(\boldsymbol{x}), y) := \phi_\gamma \big\{ \underbrace{M(f_{\boldsymbol{W}}(\boldsymbol{x}), y)}_{margin} - 2 \underbrace{\max_{k \in [K]} \epsilon \left\| W_{k,:}^L \right\|_1 \Pi_{m=1}^{N-1} \|\boldsymbol{W}^m\|_{1,\infty} \Pi_{k=1}^{L-N-1} \left\| (\boldsymbol{W}^{L-k})^T \right\|_{1,\infty} \|\boldsymbol{x}\|_1}_{worst\text{-}case\ error} \big\}$$

*Then we have upper and lower bounds of $\hat{\ell}$ in terms of 0-1 losses expressed as*

$$\max_{\hat{W}^N \in \mathbb{B}_{\boldsymbol{W}^N}^\infty(\epsilon)} \mathbb{1}\{y \neq \arg\max_{y' \in [K]} [f_{\hat{\boldsymbol{W}}}(\boldsymbol{x})]_{y'}\} \leq \hat{\ell}(f_{\boldsymbol{W}}(\boldsymbol{x}), y) \leq$$
$$\mathbb{1}\{M(f_{\boldsymbol{W}}(\boldsymbol{x}), y) - 2\max_{k \in [K]} \epsilon \left\| W_{k,:}^L \right\|_1 \Pi_{m=1}^{N-1} \|\boldsymbol{W}^m\|_{1,\infty} \Pi_{k=1}^{L-N-1} \left\| (\boldsymbol{W}^{L-k})^T \right\|_{1,\infty} \|\boldsymbol{x}\|_1 \leq \gamma\}.$$

*Proof*: Please see Appendix B.1

One could observe in the formula that the margin function $M(f_{\boldsymbol{W}}(\mathbf{x}), y)$ serves as an accuracy objective similar to the standard training process while the latter term could be conceived as the worst-case error caused by weight perturbation that should be suppressed. Therefore, by training under such an objective, we can simulate the scenario of robust training. Another intuition on the surrogate loss function is that the surrogate loss also implies the difficulty of training robust and generalizable models against large weight perturbations. Since error caused by perturbations would be surging rapidly through layers, only small perturbations can be applied in training and practice, permitting the worst-case error term to be smaller than the margin term. However, one follow-up question that naturally arises is whether or not the generalization gap will be widened when training with the robust surrogate loss. The following section investigates the generalization property while conducting robust training and provides some theoretical insights to training toward a generalizable and robust model under weight perturbation.

### 3.4.2 GENERALIZATION GAP FOR ROBUST SURROGATE LOSS

We consider the robust surrogate loss established in Lemma 2 and study its generalization bound via Rademacher complexity in Theorem 4, where $\mathcal{S} = \{(\mathbf{x}_i, y_i)\}_{i=1}^n$ denotes the set of i.i.d training samples, $\mathbf{X} := [\mathbf{x}_1, \mathbf{x}_2, \ldots, \mathbf{x}_n] \in \mathbb{R}^{d \times n}$ denotes the matrix composed of training data samples, and $d_{\max} = \max\{d, d_1, \ldots, d_L\}$ denotes the maximum dimension among all weight matrices.

**Theorem 4 (generalization gap for robust surrogate loss)** *With Lemma 2, consider the neural network hypothesis class $\mathcal{F} = \{f_{\boldsymbol{W}}(\boldsymbol{x}) | \boldsymbol{W} = (\boldsymbol{W}^1, \boldsymbol{W}^2, ..., \boldsymbol{W}^L), \|\boldsymbol{W}^h\|_\sigma \leq s_h, \|(\boldsymbol{W}^h)^T\|_{2,1} \leq b_h, h \in [L]\}$. For any $\gamma > 0$, with probability at least $1 - \delta$, we have for all $f_{\boldsymbol{W}}(\cdot) \in \mathcal{F}$*

$$\mathbb{P}_{(x,y) \sim D}\big\{\exists \hat{\boldsymbol{W}}^N \in \mathbb{B}_{\boldsymbol{W}^N}^\infty(\epsilon) \ s.t. \ y \neq \arg\max_{y' \in [K]} [f_{\hat{\boldsymbol{W}}}(\boldsymbol{x})]_{y'}\big\} \leq \frac{1}{n} \sum_{i=1}^n \mathbb{1}\Big([f_{\boldsymbol{W}}(\boldsymbol{x}_i)]_{y_i} \leq \gamma +$$

$$\max_{y' \neq y_i} [f_{\boldsymbol{W}}(\boldsymbol{x}_i)]_{y'} + 2\max_{k \in [K]} \epsilon \left\| W_{k,:}^L \right\|_1 \Pi_{m=1}^{N-1} \|\boldsymbol{W}^m\|_{1,\infty} \Pi_{k=1}^{L-N-1} \left\| (\boldsymbol{W}^{L-k})^T \right\|_{1,\infty} \|\boldsymbol{x}_i\|_1 \Big) +$$

$$\frac{1}{\gamma} \Big( \frac{4}{n^{3/2}} + \frac{60 \log(n) \log(2d_{max})}{n} \|\boldsymbol{X}\|_F \Big( \Pi_{h=1}^L s_h \Big) \Big( \sum_{j=1}^L \big( \frac{b_j}{s_j} \big)^{2/3} \Big)^{3/2} +$$

$$\underbrace{\frac{2\epsilon \sup_{f \in \mathcal{F}} \Pi_{m=1}^{N-1} \|\boldsymbol{W}^m\|_{1,\infty} \Pi_{k=0}^{L-N-1} \left\| (\boldsymbol{W}^{L-k})^T \right\|_{1,\infty}}{n} \|\boldsymbol{X}\|_{1,2} \Big) + 3\sqrt{\frac{\log \frac{2}{\delta}}{2n}}}_{complexity\ term\ caused\ by\ robust\ training}$$

*Proof*: Please see Appendix C.1

As highlighted in the bracket term of Theorem 4, if the product of multiple weight norm bounds is not well confined, the model will suffer from a notable generalization gap. Consequently, our analysis suggests a solution to reduce the generalization gap by imposing norm penalty functions on all weight matrices for training generalizable neural networks subject to weight perturbations.

### 3.5 THEORY-DRIVEN LOSS TOWARD ROBUSTNESS AND GENERALIZATION

With our theoretical insights, we now propose a robust and generalizable loss function. Standard neural network classifier training uses a classification loss $\ell_{cls}(f_{\boldsymbol{W}}(\mathbf{x}), y)$ that aims to widen the pairwise margin so as to raise accuracy, but won't necessarily be able to curb the error in output once weight perturbation is imposed. To address this issue, we propose to train under a mixed and regularized objective given a data sample $(\mathbf{x}, y)$, which would, in turn, balance the tradeoff between standard accuracy, robustness, and generalization. The designed loss takes the form:

$$\ell^*(f_{\boldsymbol{W}}(\mathbf{x}), y) = \underbrace{\ell_{cls}(f_{\boldsymbol{W}}(\mathbf{x}), y)}_{\text{standard loss}} + \lambda \cdot \underbrace{\max_{y' \neq y}\{\eta_{\boldsymbol{W}}^{y'y}(\mathbf{x}|I)\}}_{\text{robustness loss from Thm. 3}} + \mu \cdot \underbrace{\sum_{m=1}^{L} \left( \left\| (\boldsymbol{W}^m)^T \right\|_{1,\infty} + \left\| \boldsymbol{W}^m \right\|_{1,\infty} \right)}_{\text{generalization gap regularization from Thm. 4}} \quad (9)$$

The first term in the proposed loss function originates from standard classification problem for task-specific accuracy, while the second term results from the maximum error on pairwise margin (the term $\eta_{\boldsymbol{W}}^{y'y}(x|I)$ defined in Theorem 3) induced by weight perturbation. Finally, we contribute the last term to the theoretical findings in Theorem 4, where imposing norm constraints on weight matrices could benefit generalization and prevent the generalization gap from widening.

## 4 NUMERICAL VALIDATION

We validate our theoretical results and the designed loss function in (9) through two sets of experiments: *empirical generalization gap with matrix norm regularization* and *robust accuracy against adversarial weight perturbations*.

**Experiment setup**   We used the MNIST dataset comprised of gray-scale images of hand-written digits with ten categories. We trained neural network models as in (3) with four dense layers (number of neurons are 128-64-32-10) and the ReLU activation function without the bias term. We used the loss function $\ell^*$ in (9) with all-layer perturbation bound (i.e., $I = [L]$), identical weight perturbation radius $\epsilon$ (or $\epsilon_{\text{train}}$), cross entropy as the standard classification loss $\ell_{cls}$, and a batch size of 32 with 20 epochs. Stochastic gradient descent with momentum is used for training, with the learning rate set to be 0.01. For the generalization experiment, we follow the same setting as in (Yin et al., 2019), which uses 1000 data samples to train the neural network with 100 epochs. All experiments were conducted using an Intel Xeon E5-2620v4 CPU, 125 GB RAM, and an NVIDIA TITAN Xp GPU with 12 GB RAM. For reproducibility, our codes are given in the supplementary material.

**Empirical generalization gap**   Figure 1 (a) shows the empirical generalization gap (training accuracy - test accuracy) with respect to the matrix norm regularization coefficient $\mu$ defined in (9). As indicated in Theorem 4, increasing $\mu$ effectively suppresses the Rademacher complexity and thus reduces the generalization gaps, which are consistently observed on neural networks trained with different weight perturbation level $\epsilon$.

**Robustness against adversarial weight perturbation**   To evaluate the robustness against weight perturbations, we modified the projected gradient descent (PGD) attack originally designed for input perturbation (Madry et al., 2018), which we call as weight PGD attack. Starting from a trained neural network weight $\boldsymbol{W}$, the perturbed weight $\widetilde{\boldsymbol{W}}$ is crafted by iterative gradient ascent using the signed gradient of the standard loss denoted as $\text{sgn}(\nabla_{\boldsymbol{W}}\ell_{cls}(f_{\widetilde{\boldsymbol{W}}}(\mathbf{x}), y))$, followed by an element-wise $\epsilon$ clipping centered at $\boldsymbol{W}$. The attack iteration with the step size $\alpha$ is expressed as

$$\widetilde{\boldsymbol{W}}^{(0)} = \boldsymbol{W}, \ \widetilde{\boldsymbol{W}}^{(t+1)} = \text{Clip}_{\boldsymbol{W}, \epsilon}\left\{ \widetilde{\boldsymbol{W}}^{(t)} + \alpha \ \text{sgn}(\nabla_{\boldsymbol{W}}\ell_{cls}(f_{\widetilde{\boldsymbol{W}}^{(t)}}(\mathbf{x}), y)) \right\} \quad (10)$$

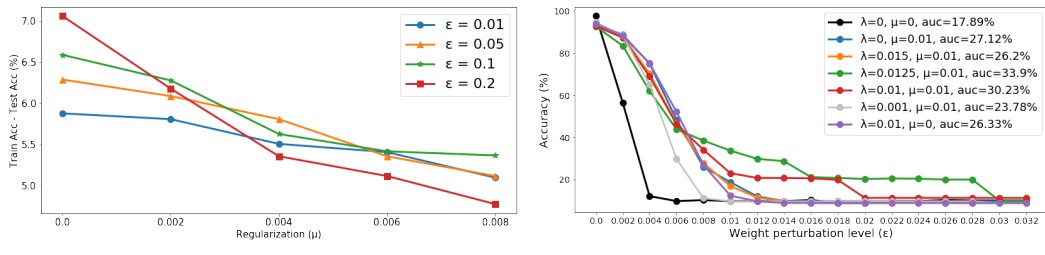

(a) Empirical generalization gap  (b) Robust accuracy under adversarial perturbation

Figure 1: (a) Empirical generalization gaps when varying the matrix norm regularization coefficient $\mu$ in (9). Consistent with the theoretical results, the gap reduces as $\mu$ increases for every $\epsilon$ value used for training. (b) Comparison of test accuracy of neural networks trained with different coefficients $\lambda$ and $\mu$ under weight PGD attack (200 steps) with $\epsilon$ perturbation level. AUC refers to the area under curve score. Joint training with the two theory-driven terms as described in (9) indeed yields more generalizable and robust neural networks against weight perturbations.

We trained neural networks with different combinations of the coefficients $\lambda$ and $\mu$ in (9) using $\epsilon_{\text{train}} = 0.01$. Figure 1 (b) shows the test accuracy under different weight perturbation level $\epsilon$ (i.e., the robust accuracy) with 200 attack steps. The standard model ($\lambda = \mu = 0$) is fragile to weight PGD attack. On the other hand, neural networks trained only with the robustness loss ($\lambda > 0$ and $\mu = 0$) or the generalization gap regularization ($\lambda = 0$ and $\mu > 0$) can improve the robust accuracy due to improved generalization and classification margin. Moreover, joint training using the proposed loss with proper coefficients can further boost model performance (e.g., $(\lambda, \mu) = (0.0125, 0.01)$), as seen by the significantly improved area under curve (AUC) score of robust accuracy over all tested $\epsilon$ values. The AUC of the best model is about $2\times$ larger than that of the standard model. Similar results can be concluded for the attack with 100 steps (see Appendix D). In Appendix D.3, we conduct additional experiments on the coefficients $\lambda$ and $\mu$ and discuss their tradeoffs. We also report the run-time analysis in Appendix F. Training with our loss function has a comparable per-epoch run time when comparing to standard training.

## 5 CONCLUSION

In this paper, we developed a formal analysis of the robustness of the pairwise class margin for neural networks against weight perturbations. We also characterized its generalization gap through Rademacher complexity. A theory-driven loss function was proposed, and the empirical results showed significantly improved performance in generalization and robustness. Our analysis offers theoretical insights and training loss design principles for studying the generalization and robustness of neural networks subject to weight perturbations.

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

# A MARGIN BOUND

## A.1 SINGLE-LAYER BOUND

We shall first prove when $N \neq L$ and follow similar reasoning to prove the case when $N = L$. Consider the difference between set of pairwise margin $f_{\widehat{\boldsymbol{W}}}^{ij}(x) - f_{\boldsymbol{W}}^{ij}(x)$, we have

$$
f_{\widehat{\boldsymbol{W}}}^{ij}(\mathbf{x}) - f_{\boldsymbol{W}}^{ij}(\mathbf{x})
$$
$$
= \{W_{i,:}^L - W_{j,:}^L\}(\hat{\mathbf{z}}^{L-1} - \mathbf{z}^{L-1}) \tag{11}
$$
$$
\overset{(a)}{\leq} \left\| W_{i,:}^L - W_{j,:}^L \right\|_1 \left\| \rho(\mathbf{W}^{L-1}\hat{\mathbf{z}}^{L-2}) - \rho(\mathbf{W}^{L-1}\mathbf{z}^{L-2}) \right\|_\infty \tag{12}
$$
$$
\overset{(b)}{\leq} \left\| W_{i,:}^L - W_{j,:}^L \right\|_1 \left\| \mathbf{W}^{L-1}(\hat{\mathbf{z}}^{L-2} - \mathbf{z}^{L-2}) \right\|_\infty \tag{13}
$$
$$
\overset{(c)}{\leq} \left\| W_{i,:}^L - W_{j,:}^L \right\|_1 \left\| (\mathbf{W}^{L-1})^T \right\|_{1,\infty} \left\| (\hat{\mathbf{z}}^{L-2} - \mathbf{z}^{L-2}) \right\|_\infty \tag{14}
$$
$$
\overset{(d)}{\leq} \left\| W_{i,:}^L - W_{j,:}^L \right\|_1 \left\| (\mathbf{W}^{L-1})^T \right\|_{1,\infty} \cdots \left\| (\mathbf{W}^{N+1})^T \right\|_{1,\infty} \left\| (\hat{\mathbf{W}}^N - \mathbf{W}^N)\mathbf{z}^{N-1} \right\|_\infty \tag{15}
$$
$$
\overset{(e)}{\leq} \epsilon \left\| W_{i,:}^L - W_{j,:}^L \right\|_1 \left\| \mathbf{z}^{N-1} \right\|_1 \Pi_{k=1}^{L-N-1} \left\| (\mathbf{W}^{L-k})^T \right\|_{1,\infty}, \tag{16}
$$

where inequality (a) results from applying Hölder inequality, and inequality (b) comes from the contractive property (1-Lipschitz) of activation function $\rho(\cdot)$. Inequality (c) and (d) come from triangle inequality applied element-wise on vector $\mathbf{W}^{L-1}(\hat{\mathbf{z}}^{L-2} - \mathbf{z}^{L-2})$ combined with induction while inequality (e) simply comes from the fact that every element in matrix $\hat{\mathbf{W}}^N - \mathbf{W}^N$ has at most $\epsilon$ in magnitude.

With similar reasoning, we proof the scenario when $N = L$ as following

$$
f_{\widehat{\boldsymbol{W}}}^{ij}(\mathbf{x}) - f_{\boldsymbol{W}}^{ij}(\mathbf{x})
$$
$$
= \{W_{i,:}^L - W_{j,:}^L\}\mathbf{z}^{L-1} - \{\hat{W}_{i,:}^L - \hat{W}_{j,:}^L\}\mathbf{z}^{L-1} \tag{17}
$$
$$
\overset{(i)}{\leq} 2\epsilon \mathbf{1}^T \mathbf{z}^{L-1} \tag{18}
$$
$$
= 2\epsilon \left\| \mathbf{z}^{L-1} \right\|_1, \tag{19}
$$

where inequality $(i)$ comes from problem definition (within element-wise $\ell_\infty$ norm ball) and since the activation function $\rho(\cdot)$ is non-negative, we could transform the inner product to its $\ell_1$ norm.

## A.2 MULTI-LAYER SCENARIO

### A.2.1 KEY PREREQUISITE

Before going through the proof for all-perturbed bound, we shall be introducing a maximization problem and search for its solution. Recall that we have defined in Section 3.1 the notion of output vector under weight perturbation setting, we now consider maximizing its $\ell_1$ norm over a given perturbed matrix set $\widehat{W}$ and give the following solution. Using the notation in Section 3.1, we have $\hat{\mathbf{z}}^k$ as the output vector under perturbation setting and write the optimal vector that achieve its maximum $\ell_1$ norm as $\mathbf{z}^{k^*}$. We then obtain the following solution

$$\left\|\mathbf{z}^{k^*}\right\|_1 = \max_{\widehat{W}} \left\|\hat{\mathbf{z}}^k\right\|_1,$$

$$\text{where } \mathbf{z}^{k^*} = \rho(\mathbf{W}^{k^*}...\rho(\mathbf{W}^{1^*}\mathbf{x}) \text{ with } \begin{cases} \mathbf{W}_{i,j}^{m^*} = W_{i,j}^m + \epsilon_m, \ \forall i,j \ \forall m \in \{2,...,L\} \\ \mathbf{W}_{i,j}^{1^*} = W_{i,j}^1 + sgn([\mathbf{x}]_j)\epsilon_1, \ \forall i,j \end{cases}$$

The reasoning behind uses the non-negative property of activation function, for any element in a matrix $\mathbf{W}$, the choice to maximize it's $\ell_1$ norm matrix-vector product is to go in the direction identical to the sign of the vector's element-wise value since the activation function is applied after the first layer, we would then obtain the solution above.

### A.2.2 ALL-PERTURBED BOUND

In the following proof for Theorem 2, we apply similar steps in Appendix A.1 and consider the difference between set of pairwise margin under natural and weight perturbation setting, recall in Theorem 2 we defined that $f_W(\mathbf{x}) = \mathbf{W}^L(..\mathbf{W}^N...\rho(\mathbf{W}^1\mathbf{x})...)$ and $f_{\widehat{W}}(\mathbf{x}) = \hat{\mathbf{W}}^L(..\hat{\mathbf{W}}^N...\rho(\hat{\mathbf{W}}^1\mathbf{x})...)$ Thus for any set of pairwise margin $f_{\widehat{W}}^{ij}(\mathbf{x})$ and $f_W^{ij}(\mathbf{x})$, we have

$$f_{\widehat{W}}^{ij}(\mathbf{x}) - f_W^{ij}(\mathbf{x})$$
$$= \{\hat{W}_{i,:}^L - \hat{W}_{j,:}^L\}\hat{\mathbf{z}}^{L-1} - \{W_{i,:}^L - W_{j,:}^L\}\mathbf{z}^{L-1} \tag{20}$$

$$\overset{(a)}{\leq} \left\|W_{i,:}^L - W_{j,:}^L\right\|_1 \left\|\rho(\hat{\mathbf{W}}^{L-1}\hat{\mathbf{z}}^{L-2}) - \rho(\mathbf{W}^{L-1}\mathbf{z}^{L-2})\right\|_\infty + 2\epsilon_L \mathbf{1}^T\hat{\mathbf{z}}^{L-1} \tag{21}$$

$$\overset{(b)}{\leq} \left\|W_{i,:}^L - W_{j,:}^L\right\|_1 \left\{ \left\|\mathbf{W}^{L-1}(\hat{\mathbf{z}}^{L-2} - \mathbf{z}^{L-2})\right\|_\infty + \left\|(\hat{\mathbf{W}}^{L-1} - \mathbf{W}^{L-1})\hat{\mathbf{z}}^{L-2}\right\|_\infty \right\} + 2\epsilon_L \left\|\hat{\mathbf{z}}^{L-1}\right\|_1 \tag{22}$$

$$\overset{(c)}{\leq} \left\|W_{i,:}^L - W_{j,:}^L\right\|_1 \left\{ \left\|(\mathbf{W}^{L-1})^T\right\|_{1,\infty} \left\|\rho(\hat{\mathbf{W}}^{L-2}\hat{\mathbf{z}}^{L-3}) - \rho(\mathbf{W}^{L-2}\mathbf{z}^{L-3})\right\|_\infty + \epsilon_{L-1}\left\|\hat{\mathbf{z}}^{L-2}\right\|_1 \right\}$$
$$+ 2\epsilon_L \left\|\hat{\mathbf{z}}^{L-1}\right\|_1 \tag{23}$$

$$\overset{(d)}{\leq} \left\|W_{i,:}^L - W_{j,:}^L\right\|_1 \left\{ \epsilon_1 \left\|\mathbf{x}\right\|_1 \Pi_{l=1}^{L-2}\left\|(\mathbf{W}^{L-l})^T\right\|_{1,\infty} + \sum_{j=1}^{L-3}\left(\Pi_{k=j+2}^{L-1}\left\|(\mathbf{W}^k)^T\right\|_{1,\infty}\right)\epsilon_{j+1}\left\|\hat{\mathbf{z}}^j\right\|_1 \right.$$
$$\left. + \epsilon_{L-1}\left\|\hat{\mathbf{z}}^{L-2}\right\|_1 \right\} + 2\epsilon_L\left\|\hat{\mathbf{z}}^{L-1}\right\|_1 \tag{24}$$

$$\overset{(e)}{\leq} \left\|W_{i,:}^L - W_{j,:}^L\right\|_1 \left\{ \epsilon_1 \left\|\mathbf{x}\right\|_1 \Pi_{l=1}^{L-2}\left\|(\mathbf{W}^{L-l})^T\right\|_{1,\infty} + \sum_{j=1}^{L-3}\left(\Pi_{k=j+2}^{L-1}\left\|(\mathbf{W}^k)^T\right\|_{1,\infty}\right)\epsilon_{j+1}\left\|\mathbf{z}^{j^*}\right\|_1 \right.$$
$$\left. + \epsilon_{L-1}\left\|\mathbf{z}^{L-2^*}\right\|_1 \right\} + 2\epsilon_L\left\|\mathbf{z}^{L-1^*}\right\|_1 \tag{25}$$

In the above proof, inequality (a) comes from the problem definition (perturbation of final layer within $\epsilon_L$) and inequality (b) results from the contractive property of $\rho(\cdot)$ (1-Lipschitz) combined with the use of triangle inequality. Inequality (c) was achieved through triangle inequality applied on elements of $\mathbf{W}^{L-1}(\hat{\mathbf{z}}^{L-2} - \mathbf{z}^{L-2})$ and using the fact that $\hat{\mathbf{W}}^{L-1} - \mathbf{W}^{L-1}$ has every element less

than or equal to $\epsilon_{L-1}$ in magnitude. By the process of induction and maximizing the $\ell_1$ norm of perturbed output under weight perturbation $\hat{\mathbf{z}}^k$, we arrive at inequality (d) and (e).

### A.2.3 MULTI-LAYER BOUND

We now proceed to utilize similar reasoning to establish the multi-layer bound when weight perturbation is imposed according to an index set $I$

$$
\begin{aligned}
& f_{\hat{\boldsymbol{W}}}^{ij}(\mathbf{x}) - f_{\boldsymbol{W}}^{ij}(\mathbf{x}) \\
&= \{\tilde{W}_{i,:}^L - \tilde{W}_{j,:}^L\}\hat{\mathbf{z}}^{L-1} - \{W_{i,:}^L - W_{j,:}^L\}\mathbf{z}^{L-1} && (26) \\
&\leq \left\|W_{i,:}^L - W_{j,:}^L\right\|_1 \left\|\rho(\tilde{\mathbf{W}}^{L-1}\hat{\mathbf{z}}^{L-2}) - \rho(\mathbf{W}^{L-1}\mathbf{z}^{L-2})\right\|_\infty + \mathbb{1}(L \in I)2\epsilon_L\mathbf{1}^T\hat{\mathbf{z}}^{L-1} && (27) \\
&\leq \left\|W_{i,:}^L - W_{j,:}^L\right\|_1 \Big\{ \left\|\mathbf{W}^{L-1}(\hat{\mathbf{z}}^{L-2} - \mathbf{z}^{L-2})\right\|_\infty + \left\|(\tilde{\mathbf{W}}^{L-1} - \mathbf{W}^{L-1})\hat{\mathbf{z}}^{L-2}\right\|_\infty \Big\} \\
&\quad + \mathbb{1}(L \in I)2\epsilon_L \left\|\hat{\mathbf{z}}^{L-1}\right\|_1 && (28) \\
&\leq \left\|W_{i,:}^L - W_{j,:}^L\right\|_1 \Big\{ \left\|(\mathbf{W}^{L-1})^T\right\|_{1,\infty} \left\|\rho(\tilde{\mathbf{W}}^{L-2}\hat{\mathbf{z}}^{L-3}) - \rho(\mathbf{W}^{L-2}\mathbf{z}^{L-3})\right\|_\infty \\
&\quad + \mathbb{1}(L-1 \in I)\epsilon_{L-1}\left\|\hat{\mathbf{z}}^{L-2}\right\|_1 \Big\} + \mathbb{1}(L \in I)2\epsilon_L\left\|\hat{\mathbf{z}}^{L-1}\right\|_1 && (29) \\
&\leq \left\|W_{i,:}^L - W_{j,:}^L\right\|_1 \Big\{ \mathbb{1}(1 \in I)\epsilon_1 \left\|\mathbf{x}\right\|_1 \Pi_{l=1}^{L-2}\left\|(\mathbf{W}^{L-l})^T\right\|_{1,\infty} + \mathbb{1}(L-1 \in I)\epsilon_{L-1}\left\|\hat{\mathbf{z}}^{L-2}\right\|_1 \\
&\quad + \sum_{j=1}^{L-3} \mathbb{1}(j+1 \in I)\big(\Pi_{k=j+2}^{L-1}\left\|(\mathbf{W}^k)^T\right\|_{1,\infty}\big)\epsilon_{j+1}\left\|\hat{\mathbf{z}}^j\right\|_1 \Big\} + \mathbb{1}(L \in I)2\epsilon_L\left\|\hat{\mathbf{z}}^{L-1}\right\|_1 && (30) \\
&\leq \left\|W_{i,:}^L - W_{j,:}^L\right\|_1 \Big\{ \sum_{\ell \in I\backslash\{L,L-1\}} \big(\Pi_{k=\ell+1}^{L-1}\left\|(\mathbf{W}^k)^T\right\|_{1,\infty}\big)\epsilon_\ell\left\|\mathbf{z}^{\ell-1^*}\right\|_1 \\
&\quad + \mathbb{1}(L-1 \in I)\epsilon_{L-1}\left\|\mathbf{z}^{L-2^*}\right\|_1 \Big\} + \mathbb{1}(L \in I)2\epsilon_L\left\|\mathbf{z}^{L-1^*}\right\|_1 && (31)
\end{aligned}
$$

The proof for multi-layer bound follows same reasoning from the all-perturbed setting except indicator function was added to check whether a certain layer $m$ is in the index set $I$ and at last we rewrite the expression using the members of set $I$.

## B SURROGATE LOSS

### B.1 CASE ON RAMP LOSS

We now provide a proof for Lemma 2. Recall the definition of ramp function in Section 3.4.1, we have that ramp loss for a given data point $(\mathbf{x}, y)$ and neural network $f_{\boldsymbol{W}}(\cdot)$ is written as $\ell_{\text{ramp}}(f_{\boldsymbol{W}}(\mathbf{x}), y) = \phi_\gamma(M(f_{\boldsymbol{W}}(\mathbf{x}), y))$, where the function $\phi_\gamma : \mathbb{R} \mapsto [0, 1]$ is defined as

$$
\phi_\gamma(t) = \begin{cases} 1 & if \quad t \leq 0 \\ 0 & if \quad t \geq \gamma \\ 1 - \frac{t}{\gamma} & if \quad t \in [0, \gamma] \end{cases} \tag{32}
$$

Then for any $(\mathbf{x}, y)$, using ReLU as activation function, we have

$$\max_{\widehat{\boldsymbol{W}}} \mathbb{1}(y \neq \arg\max_{y'}[f_{\widehat{\boldsymbol{W}}}(\mathbf{x})]_{y'})$$

$$\overset{(a)}{\leq} \phi_\gamma(\min_{\widehat{\boldsymbol{W}}} M(f_{\widehat{\boldsymbol{W}}}(\mathbf{x}), y)) \tag{33}$$

$$\overset{(b)}{\leq} \phi_\gamma(\min_{y'\neq y}\min_{\widehat{\boldsymbol{W}}}[f_{\widehat{\boldsymbol{W}}}(\mathbf{x})]_y - [f_{\widehat{\boldsymbol{W}}}(\mathbf{x})]_{y'}) \tag{34}$$

$$\overset{(c)}{\leq} \phi_\gamma\big(\min_{y'\neq y}[f_{\boldsymbol{W}}(\mathbf{x})]_y - [f_{\boldsymbol{W}}(\mathbf{x})]_{y'} - \max_{y'\neq y}\epsilon\left\|W_{y',:}^L - W_{y,:}^L\right\|_1 \left\|\mathbf{z}^{N-1}\right\|_1 \Pi_{k=1}^{L-N-1}\left\|(\mathbf{W}^{L-k})^T\right\|_{1,\infty}\big) \tag{35}$$

$$\overset{(d)}{\leq} \phi_\gamma\big(\min_{y'\neq y}[f_{\boldsymbol{W}}(\mathbf{x})]_y - [f_{\boldsymbol{W}}(\mathbf{x})]_{y'} - 2\max_{k\in[K]}\epsilon\left\|W_{k,:}^L\right\|_1 \left\|\mathbf{z}^{N-1}\right\|_1 \Pi_{k=1}^{L-N-1}\left\|(\mathbf{W}^{L-k})^T\right\|_{1,\infty}\big) \tag{36}$$

$$\overset{(e)}{\leq} \phi_\gamma\big(M(f_{\boldsymbol{W}}(\mathbf{x}), y) - 2\max_{k\in[K]}\epsilon\left\|W_{k,:}^L\right\|_1 \left\|\mathbf{z}^{N-1}\right\|_1 \Pi_{k=1}^{L-N-1}\left\|(\mathbf{W}^{L-k})^T\right\|_{1,\infty}\big) \tag{37}$$

$$\overset{(f)}{\leq} \phi_\gamma\big(M(f_{\boldsymbol{W}}(\mathbf{x}), y) - 2\max_{k\in[K]}\epsilon\left\|W_{k,:}^L\right\|_1 \left\|\mathbf{x}\right\|_1 \Pi_{m=1}^{N-1}\left\|\mathbf{W}^m\right\|_{1,\infty}\Pi_{k=1}^{L-N-1}\left\|(\mathbf{W}^{L-k})^T\right\|_{1,\infty}\big) \tag{38}$$

$$:= \hat{\ell}(f_{\boldsymbol{W}}(\mathbf{x}), y) \tag{39}$$

$$\overset{(g)}{\leq} \mathbb{1}\big(M(f_{\boldsymbol{W}}(\mathbf{x}), y) - 2\max_{k\in[K]}\epsilon\left\|W_{k,:}^L\right\|_1 \left\|\mathbf{x}\right\|_1 \Pi_{m=1}^{N-1}\left\|\mathbf{W}^m\right\|_{1,\infty}\Pi_{k=1}^{L-N-1}\left\|(\mathbf{W}^{L-k})^T\right\|_{1,\infty} \leq \gamma\big), \tag{40}$$

where inequality (a) is due to the property of ramp loss while inequality (b) is by the definition of margin and inequality (c) comes from applying Theorem 1. Inequality (d) results from using triangle inequality and taking its maximum, inequality (e) is by the definition of margin and inequality (f) comes from the fact that with ReLU we have $\|\rho(\mathbf{Ax})\|_1 \leq \|\mathbf{Ax}\|_1$. Lastly, inequality (g) is a direct consequence from property of ramp loss.

### B.1.1 RAMP LOSS ON MULTIPLE LAYER BOUND

We now follow a similar course and prove robust ramp loss using the multi-layer bound in Theorem 3. We consider the robust loss form proposed in Section 3.4.1 and have that,

$$\max_{\widehat{\boldsymbol{W}}} \ell_{\text{ramp}}(f_{\widehat{\boldsymbol{W}}}(\mathbf{x}), y)$$

$$\overset{(a)}{\leq} \phi_\gamma(\min_{\widehat{\boldsymbol{W}}} M(f_{\widehat{\boldsymbol{W}}}(\mathbf{x}), y)) \tag{41}$$

$$\overset{(b)}{\leq} \phi_\gamma\big(\min_{y'\neq y}[f_{\boldsymbol{W}}(\mathbf{x})]_y - [f_{\boldsymbol{W}}(\mathbf{x})]_{y'} - \max_{y'\neq y}\eta_{\boldsymbol{W}}^{y'y}(\mathbf{x}|I)\big) \tag{42}$$

$$\overset{(c)}{\leq} \phi_\gamma\big(M(f_{\boldsymbol{W}}(\mathbf{x}), y) - \max_{y'\neq y}\eta_{\boldsymbol{W}}^{y'y}(\mathbf{x}|I)\big) := \hat{\ell}(f_{\boldsymbol{W}}(\mathbf{x}), y) \tag{43}$$

### B.2 CROSS ENTROPY

We further consider the case of cross entropy and prove an upper bound for it. We denote the loss function as $CE(\cdot)$, and during training, hard label was applied. Recall the definition of $\tilde{f}_{\boldsymbol{W}}(\mathbf{x})$ in

Section 3.4.1, we have the difference of loss function between natural and perturbation settings as,

$$CE(\tilde{f}_{\widehat{\boldsymbol{W}}}(\mathbf{x}), y) - CE(\tilde{f}_{\boldsymbol{W}}(\mathbf{x}), y)$$

$$= -y \ln \frac{[\tilde{f}_{\widehat{\boldsymbol{W}}}(\mathbf{x})]_y}{[\tilde{f}_{\boldsymbol{W}}(\mathbf{x})]_y} \tag{44}$$

$$= \ln \frac{[\tilde{f}_{\boldsymbol{W}}(\mathbf{x})]_y}{[\tilde{f}_{\widehat{\boldsymbol{W}}}(\mathbf{x})]_y} \tag{45}$$

$$= \ln \left( e^{[f_{\boldsymbol{W}}(\mathbf{x})]_y - [f_{\widehat{\boldsymbol{W}}}(\mathbf{x})]_y} \frac{\sum_{k \in [K]} e^{[f_{\widehat{\boldsymbol{W}}}(\mathbf{x})]_k}}{\sum_{k \in [K]} e^{[f_{\boldsymbol{W}}(\mathbf{x})]_k}} \} \right. \tag{46}$$

$$\overset{(a)}{\leq} \ln \left( \max_{y' \neq y} e^{[f_{\boldsymbol{W}}(\mathbf{x})]_y - [f_{\widehat{\boldsymbol{W}}}(\mathbf{x})]_y} \cdot e^{[f_{\widehat{\boldsymbol{W}}}(\mathbf{x})]_{y'} - [f_{\boldsymbol{W}}(\mathbf{x})]_{y'}} \right) \tag{47}$$

$$\overset{(b)}{\leq} \ln \left( e^{\max_{y' \neq y} f_{\widehat{\boldsymbol{W}}}^{y'y}(\mathbf{x}) - f_{\boldsymbol{W}}^{y'y}(\mathbf{x})} \right) \tag{48}$$

$$\overset{(c)}{=} \max_{y' \neq y} \eta_{\boldsymbol{W}}^{y'y}(\mathbf{x}|I), \tag{49}$$

where inequality (a) comes from taking the maximum in the set of all ratios $\left\{ \frac{e^{[f_{\widehat{\boldsymbol{W}}}(\mathbf{x})]_k}}{e^{[f_{\boldsymbol{W}}(\mathbf{x})]_k}} \right\}$ and inequality (b) comes from monotonicity of exponential function. Finally, the last expression (c) can be referred to Theorem 3. Thus with the above proof, we could establish the following robust surrogate loss for cross entropy, denoted as $\widehat{CE}(f_{\boldsymbol{W}}(\mathbf{x}), y)$ :

$$\widehat{CE}(f_{\boldsymbol{W}}(\mathbf{x}), y) = CE(f_{\boldsymbol{W}}(\mathbf{x}), y) + \max_{y' \neq y} \eta_{\boldsymbol{W}}^{y'y}(\mathbf{x}|I)$$

## C  GENERALIZATION BOUND ON RADEMACHER COMPLEXITY

### C.1  PROOF ON SINGLE LAYER BOUND

To show the Rademacher complexity and generalization gap on single layer robust surrogate loss, we first introduce a result proven in (Bartlett et al., 2017) and another classical result in statistical learning theory and proceed to give a proof on Theorem 4. Given a set $\mathcal{S} = \{(\mathbf{x}_i, y_i)\}_{i=1}^n$ of i.i.d training samples, denote $\mathbf{X} := [\mathbf{x}_1, \mathbf{x}_2, \dots, \mathbf{x}_n] \in \mathbb{R}^{d \times n}$ as the matrix composed of training data and let $d_{\max} = \max\{d, d_1, \dots, d_L\}$ as the maximum dimension among all weight matrices.

**Lemma 3 (Mohri et al. (2018))** *Assume that the range of loss function $\ell(\cdot)$ is [0,1]. Then, for any $\delta \in (0, 1)$, with probability as least $1 - \delta$, we have for all $f \in \mathcal{F}$*

$$R(f) \leq R_n(f) + 2\mathcal{R}(\ell_{\mathcal{F}}) + 3\sqrt{\frac{\log \frac{2}{\delta}}{2n}},$$

*where $R(f)$ and $R_n(f)$ stand for population risk and empirical risk, respectively.*

**Lemma 4 (Bartlett et al. (2017))** *Consider the neural network hypothesis class,*

$$\mathcal{F} = \{f_{\boldsymbol{W}}(\boldsymbol{x}) \mid \boldsymbol{W} = (\boldsymbol{W}^1, \boldsymbol{W}^2, ..., \boldsymbol{W}^L), \left\| \boldsymbol{W}^h \right\|_{\sigma} \leq s_h, \left\| (\boldsymbol{W}^h)^T \right\|_{2,1} \leq b_h \, h \in [L]\}$$

*We have an upper bound on the Rademacher complexity,*

$$\mathcal{R}(\mathcal{F}) \leq \frac{4}{n^{3/2}} + \frac{26 \log(n) \log(2d_{max})}{n} \|\boldsymbol{X}\|_F \left( \Pi_{h=1}^L s_h \right) \left( \sum_{j=1}^L \left( \frac{b_j}{s_j} \right)^{2/3} \right)^{3/2}$$

We now study the Rademacher Complexity of the function class

$$\hat{\ell}_{\mathcal{F}} = \{(\mathbf{x}, y) \mapsto \hat{\ell}(f_{\boldsymbol{W}}(\mathbf{x}), y) \mid f \in \mathcal{F}\},$$

where $\hat{\ell}(\cdot)$ is denoted in Lemma 2 and let $M_{\mathcal{F}} = \{(\mathbf{x}, y) \mapsto M(f_{\boldsymbol{W}}(\mathbf{x}), y) | f \in \mathcal{F}\}$. Then we could obtain,

$$\mathcal{R}(\hat{\ell}_{\mathcal{F}}) \leq \frac{1}{\gamma}\Big(\mathcal{R}(M_{\mathcal{F}}) + \frac{2\epsilon}{n}E_{\nu}[\sup_{f \in \mathcal{F}}\sum_{i=1}^{n}\nu_i \max_{k \in [K]}\big\|W_{k,:}^{L}\big\|_1 \Pi_{m=1}^{N-1}\|\mathbf{W}^m\|_1 \Pi_{k=1}^{L-N-1}\big\|(\mathbf{W}^{L-k})^T\big\|_{1,\infty}\|\mathbf{x}_i\|_1]\Big),$$
(50)

where the inequality was achieved by using the Ledoux-Talagrand contraction inequality and the convexity of the supreme operation. Consider the second term, we have that

$$\frac{2\epsilon}{n}E_{\nu}[\sup_{f \in \mathcal{F}}\sum_{i=1}^{n}\nu_i \max_{k \in [K]}\big\|W_{k,:}^{L}\big\|_1 \Pi_{m=1}^{N-1}\|\mathbf{W}^m\|_1 \Pi_{k=1}^{L-N-1}\big\|(\mathbf{W}^{L-k})^T\big\|_{1,\infty}\|\mathbf{x}_i\|_1]$$
(51)

$$\overset{(a)}{\leq} \frac{2\epsilon}{n}\big(\sup_{f \in \mathcal{F}}\max_{k \in [K]}\big\|W_{k,:}^{L}\big\|_1 \Pi_{m=1}^{N-1}\|\mathbf{W}^m\|_1 \Pi_{k=1}^{L-N-1}\big\|(\mathbf{W}^{L-k})^T\big\|_{1,\infty}\big)E_{\nu}[|\sum_{i=1}^{n}\nu_i\|\mathbf{x}_i\|_1|]$$
(52)

$$\overset{(b)}{\leq} \frac{2\epsilon}{n}\big(\sup_{f \in \mathcal{F}}\Pi_{m=1}^{N-1}\|\mathbf{W}^m\|_1 \Pi_{k=0}^{L-N-1}\big\|(\mathbf{W}^{L-k})^T\big\|_{1,\infty}\big)\|\mathbf{X}\|_{1,2},$$
(53)

where inequality (a) is achieved by separating all neural network related parameters and inequality (b) is a result of applying Khintchine's inequality.

Thus, combined with Lemma 4, we have that

$$\mathcal{R}(\hat{\ell}_{\mathcal{F}}) \leq \frac{1}{\gamma}\Big(\frac{4}{n^{3/2}} + \frac{60\log(n)\log(2d_{max})}{n}\|\mathbf{X}\|_F\Big(\Pi_{h=1}^{L}s_h\Big)\Big(\sum_{j=1}^{L}\big(\frac{b_j}{s_j}\big)^{2/3}\Big)^{3/2}$$
$$+ \frac{2\epsilon}{n}\big(\sup_{f \in \mathcal{F}}\Pi_{m=1}^{N-1}\|\mathbf{W}^m\|_1 \Pi_{k=0}^{L-N-1}\big\|(\mathbf{W}^{L-k})^T\big\|_{1,\infty}\big)\|\mathbf{X}\|_{1,2}$$
(54)

Once we have calculated an upper bound for $\mathcal{R}(\hat{\ell}_{\mathcal{F}})$, then Theorem 4 is a direct consequence of Lemma 2 and 3.

## C.2 EXTENSION TO MULTIPLE LAYER BOUND

In this section, we consider the robust surrogate loss under multi-layer bound and study its Rademacher complexity, We first give the expression of the robust surrogate loss then give an result on generalization bound.

**Lemma 5** *Define the robust loss function* $\hat{\ell}(f_{\boldsymbol{W}}(\boldsymbol{x}), y)$ *as*

$$\hat{\ell}(f_{\boldsymbol{W}}(\boldsymbol{x}), y) = \phi_{\gamma}\bigg(M(f_{\boldsymbol{W}}(\boldsymbol{x}), y)$$

$$- 2\max_{k \in [K]}\big\|W_{k,:}^{L}\big\|_1\bigg\{\sum_{\ell \in I \setminus \{L, L-1\}}\epsilon_{\ell}\big(\Pi_{i=1}^{l-1}\big\|\boldsymbol{W}^{i^*}\big\|_{1,\infty}\big)\big(\Pi_{j=\ell+1}^{L-1}\big\|(\boldsymbol{W}^j)^T\big\|_{1,\infty}\big)\|\boldsymbol{x}\|_1$$

$$+ \mathbb{1}(L-1 \in I)\epsilon_{L-1}\big(\Pi_{i=1}^{L-2}\big\|\boldsymbol{W}^{i^*}\big\|_{1,\infty}\big)\|\boldsymbol{x}\|_1\bigg\} - \mathbb{1}(L \in I)2\epsilon_L\big(\Pi_{i=1}^{L-1}\big\|\boldsymbol{W}^{i^*}\big\|_{1,\infty}\big)\|\boldsymbol{x}\|_1\bigg) \quad (55)$$

*We would have that*

$$\max_{\widehat{\boldsymbol{W}}}\mathbb{1}(y \neq \arg\max_{y'}[f_{\widehat{\boldsymbol{W}}}(\boldsymbol{x})]_{y'}) \leq \hat{\ell}(f_{\boldsymbol{W}}(\boldsymbol{x}), y)$$

$$\leq \mathbb{1}\bigg(M(f_{\boldsymbol{W}}(\boldsymbol{x}), y) - 2\max_{k \in [K]}\big\|W_{k,:}^{L}\big\|_1\bigg\{\sum_{\ell \in I \setminus \{L, L-1\}}\epsilon_{\ell}\big(\Pi_{i=1}^{l-1}\big\|\boldsymbol{W}^{i^*}\big\|_{1,\infty}\big)\big(\Pi_{j=\ell+1}^{L-1}\big\|(\boldsymbol{W}^j)^T\big\|_{1,\infty}\big)\|\boldsymbol{x}\|_1$$

$$+ \mathbb{1}(L-1 \in I)\epsilon_{L-1}\big(\Pi_{i=1}^{L-2}\big\|\boldsymbol{W}^{i^*}\big\|_{1,\infty}\big)\|\boldsymbol{x}\|_1\bigg\} - \mathbb{1}(L \in I)2\epsilon_L\big(\Pi_{i=1}^{L-1}\big\|\boldsymbol{W}^{i^*}\big\|_{1,\infty}\big)\|\boldsymbol{x}\|_1 \leq \gamma\bigg)$$

Using the above loss, we could further establish an upper bound on robust surrogate loss and provide statements on generalization bound. Given the following function class

$$\hat{\ell}_{\mathcal{F}} = \{(\mathbf{x}, y) \mapsto \hat{\ell}(f_{\mathbf{W}}(\mathbf{x}), y) | f \in \mathcal{F}\}$$

We have that,

$$
\mathcal{R}(\hat{\ell}_{\mathcal{F}}) \leq \frac{1}{\gamma} \Big( \mathcal{R}(M_{\mathcal{F}}) + \frac{2}{n} E_{\nu} \Big[ \sup_{f \in \mathcal{F}} \sum_{i=1}^{n} \nu_i \max_{k \in [K]} \|W_{k,:}^L\|_1 \{ \sum_{\ell \in I \setminus \{L, L-1\}} \epsilon_\ell \big( \Pi_{i=1}^{l-1} \|\mathbf{W}^{i^*}\|_{1,\infty} \big) \times
$$

$$
\big( \Pi_{j=\ell+1}^{L-1} \|(\mathbf{W}^j)^T\|_{1,\infty} \big) \|\mathbf{x}_i\|_1 + \mathbb{1}(L-1 \in I)\epsilon_{L-1} \big( \Pi_{i=1}^{L-2} \|\mathbf{W}^{i^*}\|_{1,\infty} \big) \|\mathbf{x}_i\|_1 \} \Big]
$$

$$
+ \frac{2}{n} E_{\nu} \Big[ \sup_{f \in \mathcal{F}} \sum_{i=1}^{n} \nu_i \mathbb{1}(L \in I)\epsilon_L \big( \Pi_{i=1}^{L-1} \|\mathbf{W}^{i^*}\|_{1,\infty} \big) \|\mathbf{x}_i\|_1 \Big] \Big) \tag{56}
$$

which the second term can be bounded as,

$$
\frac{2}{n} E_{\nu} \Big[ \sup_{f \in \mathcal{F}} \sum_{i=1}^{n} \nu_i \max_{k \in [K]} \|W_{k,:}^L\|_1 \{ \sum_{\ell \in I \setminus \{L, L-1\}} \epsilon_\ell \big( \Pi_{i=1}^{l-1} \|\mathbf{W}^{i^*}\|_{1,\infty} \big) \times
$$

$$
\big( \Pi_{j=\ell+1}^{L-1} \|(\mathbf{W}^j)^T\|_{1,\infty} \big) \|\mathbf{x}_i\|_1 + \mathbb{1}(L-1 \in I)\epsilon_{L-1} \big( \Pi_{i=1}^{L-2} \|\mathbf{W}^{i^*}\|_{1,\infty} \big) \|\mathbf{x}_i\|_1 \} \Big] \tag{57}
$$

$$
\leq \frac{2}{n} \sup_{f \in \mathcal{F}} \max_{k \in [K]} \{ \|W_{k,:}^L\|_1 \{ \sum_{\ell \in I \setminus \{L, L-1\}} \epsilon_\ell \big( \Pi_{i=1}^{l-1} \|\mathbf{W}^{i^*}\|_{1,\infty} \big) \big( \Pi_{j=\ell+1}^{L-1} \|(\mathbf{W}^j)^T\|_{1,\infty} \big)
$$

$$
+ \mathbb{1}(L-1 \in I)\epsilon_{L-1} \big( \Pi_{i=1}^{L-2} \|\mathbf{W}^{i^*}\|_{1,\infty} \big) \} \} E_{\nu} [| \sum_{i=1}^{n} \nu_i \|\mathbf{x}_i\|_1 |] \tag{58}
$$

$$
\leq \frac{2}{n} \sup_{f \in \mathcal{F}} \max_{k \in [K]} \{ \|W_{k,:}^L\|_1 \{ \sum_{\ell \in I \setminus \{L, L-1\}} \epsilon_\ell \big( \Pi_{i=1}^{l-1} \|\mathbf{W}^{i^*}\|_{1,\infty} \big) \big( \Pi_{j=\ell+1}^{L-1} \|(\mathbf{W}^j)^T\|_{1,\infty} \big)
$$

$$
+ \mathbb{1}(L-1 \in I)\epsilon_{L-1} \big( \Pi_{i=1}^{L-2} \|\mathbf{W}^{i^*}\|_{1,\infty} \big) \} \} \|\mathbf{X}\|_{1,2} \tag{59}
$$

while the last term can as well be bounded as,

$$
\frac{2}{n} E_{\nu} \Big[ \sup_{f \in \mathcal{F}} \sum_{i=1}^{n} \nu_i \mathbb{1}(L \in I)\epsilon_L \big( \Pi_{i=1}^{L-1} \|\mathbf{W}^{i^*}\|_{1,\infty} \big) \|\mathbf{x}_i\|_1 \Big] \tag{60}
$$

$$
\leq \frac{2 \sup_{f \in \mathcal{F}} \mathbb{1}(L \in I)\epsilon_L \big( \Pi_{i=1}^{L-1} \|\mathbf{W}^{i^*}\|_{1,\infty} \big)}{n} E_{\nu} [| \sum_{i=1}^{n} \nu_i \|\mathbf{x}_i\|_1 |] \tag{61}
$$

$$
\leq \frac{2 \mathbb{1}(L \in I)\epsilon_L \sup_{f \in \mathcal{F}} \big( \Pi_{i=1}^{L-1} \|\mathbf{W}^{i^*}\|_{1,\infty} \big)}{n} \|\mathbf{X}\|_{1,2} \tag{62}
$$

With all of the upper bounds above and Lemma 4, 5, we have the following theorem,

**Theorem 5 (generalization gap for robust surrogate loss)** *With Lemma 5, consider the neural network hypothesis class* $\mathcal{F} = \{f_{\mathbf{W}}(\mathbf{x}) | \mathbf{W} = (\mathbf{W}^1, \mathbf{W}^2, ..., \mathbf{W}^L), \|\mathbf{W}^h\|_\sigma \leq s_h, \|(\mathbf{W}^h)^T\|_{2,1} \leq b_h, h \in [L]\}$. *For any* $\gamma > 0$, *with probability at least* $1 - \delta$, *we have for all* $f_{\mathbf{W}}(\cdot) \in \mathcal{F}$

$$
\mathbb{P}_{(\mathbf{x},y) \sim D} \{ \exists \widehat{\mathbf{W}} \text{ s.t. } y \neq \arg \max_{y' \in [K]} [f_{\widehat{\mathbf{W}}}(\mathbf{x})]_{y'} \} \leq \frac{1}{n} \sum_{i=1}^{n} \mathbb{1} \Big( [f_{\mathbf{W}}(\mathbf{x}_i)]_{y_i} \leq \gamma + \max_{y' \neq y_i} [f_{\mathbf{W}}(\mathbf{x}_i)]_{y'} + 2 \max_{k \in [K]} \|W_{k,:}^L\|_1
$$

$$
\times \{ \sum_{\ell \in I \setminus \{L, L-1\}} \epsilon_\ell \big( \Pi_{i=1}^{l-1} \|\mathbf{W}^{i^*}\|_{1,\infty} \big) \big( \Pi_{j=\ell+1}^{L-1} \|(\mathbf{W}^j)^T\|_{1,\infty} \big) + \mathbb{1}(L-1 \in I)\epsilon_{L-1} \big( \Pi_{i=1}^{L-2} \|\mathbf{W}^{i^*}\|_{1,\infty} \big) \} \|\mathbf{x}_i\|_1
$$

$$
+ 2\mathbb{1}(L \in I)\epsilon_L \big( \Pi_{i=1}^{L-1} \|\mathbf{W}^{i^*}\|_{1,\infty} \big) \|\mathbf{x}_i\|_1 \Big) + \frac{1}{\gamma} \Big( \frac{4}{n^{3/2}} + \frac{60 \log(n) \log(2d_{max})}{n} \|X\|_F
$$

$$
+ \frac{2}{n} \sup_{f \in \mathcal{F}} \max_{k \in [K]} \{ \|W_{k,:}^L\|_1 \{ \sum_{\ell \in I \setminus \{L, L-1\}} \epsilon_\ell \big( \Pi_{i=1}^{l-1} \|\mathbf{W}^{i^*}\|_{1,\infty} \big) \big( \Pi_{j=\ell+1}^{L-1} \|(\mathbf{W}^j)^T\|_{1,\infty} \big)
$$

$$
+ \mathbb{1}(L-1 \in I)\epsilon_{L-1} \big( \Pi_{i=1}^{L-2} \|\mathbf{W}^{i^*}\|_{1,\infty} \big) \} + 2\mathbb{1}(L \in I)\epsilon_L \big( \Pi_{i=1}^{L-1} \|\mathbf{W}^{i^*}\|_{1,\infty} \big) \} \|\mathbf{X}\|_{1,2} \Big) + 3\sqrt{\frac{\log \frac{2}{\delta}}{2n}} \tag{63}
$$

# D  ADDITIONAL EXPERIMENTS

## D.1  SUPPLEMENT OF FIGURE 1 (A)

For completeness, Figure 2 adds the curve of $\epsilon = 0$ (the standard generalization setting) in Figure 1 (a). The generalization gap is obviously lower than others when $\epsilon = 0$, but its trend with respect to regularization is similar to the setting when $\epsilon \neq 0$.

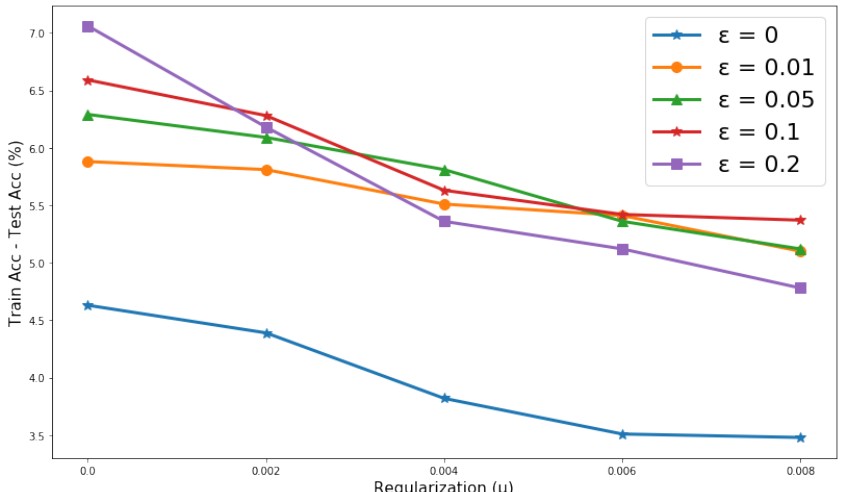

Figure 2: Empirical generalization gaps when altering the matrix norm regularization coefficient $\mu$ in (9).

## D.2  WEIGHT PGD ATTACK (100 STEPS) ON MNIST

Figure 3 shows the accuracy and AUC score of different models against weight PGD attack with 100 iterations.

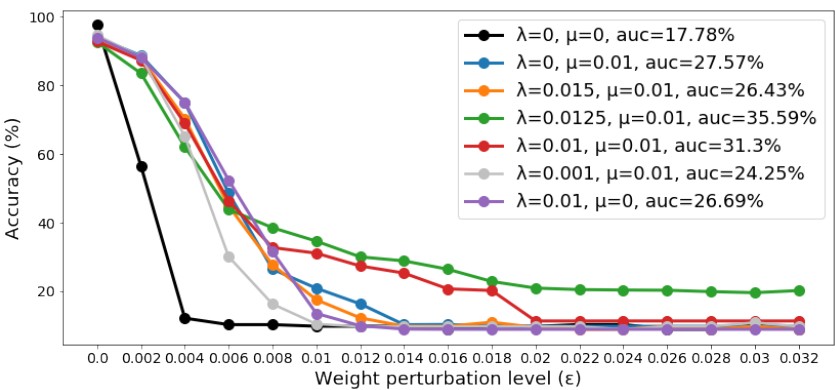

Figure 3: Comparison of test accuracy of neural networks trained with different coefficients $\lambda$ and $\mu$ against weight PGD attack (100 steps) with perturbation level $\epsilon$.

## D.3 More details on the trade-off between $\lambda$ and $\mu$

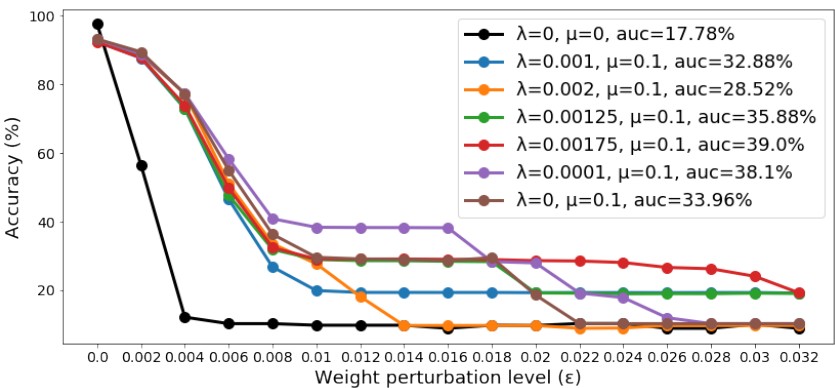

(a) weight PGD ($\epsilon = 0.01$)      (b) weight PGD ($\epsilon = 0.02$)

Figure 4: Comparison of test accuracy (%) trained with different coefficients $\lambda$ and $\mu$ against weight PGD attack (200 steps) with $\epsilon = 0.01$ and $0.02$. The experiment setting follows Figure 1(b).

Figure 4 shows the trade-off with a fine grid search between coefficients $\lambda$ and $\mu$, we present test accuracy under weight PGD attack using perturbation radius of ($\epsilon = 0.01$ and $0.02$) with different combinations of $\lambda$ (from 0.01 to 0.015) and $\mu$ (from 0 to 0.05). We find that there is indeed a sweet spot with proper values of $\lambda$ and $\mu$ leading to significantly better robust accuracy. When both $\lambda$ and $\mu$ are too large or too small, the robustness of the model will decrease.

## D.4 Alternative Robust Loss Function

In addition to the proposed loss function in equation (9), one can consider an alternative generalization gap regularization term derived from Theorem 4, which is $\sum_{m=1}^{L}(\log\,(||(\mathbf{W}^m)^\top||_{1,\infty}) + \log\,(||\mathbf{W}^m||_{1,\infty}))$. We compare its performance following the same experiment setting as in Figure 1 (b) with finetuned coefficients $\lambda$ and $\mu$. It can be observed that this alternative loss function also yields robust models with comparable (sometimes slightly better) performance to those in Figure 1 (b), verifying the effectiveness in using theory-driven insights to reduce the generalization gap against weight perturbation.

Figure 5: Test accuracy under different weight perturbation level $\epsilon$ with 200 attack steps. The models are trained using the alternative loss described in Section D.4.

# E    ON THE VACUITY OF GENERALIZATION BOUND

In (Nagarajan & Kolter, 2019), empirical observations were made to point out the fact that when given increase in the size of the training data, the error bound proposed in (Bartlett et al., 2017) grows rapidly, loosing the ability to describe generalization gap and thus becomes vacuous. However, we note that under our settings with models trained using the loss function in Section 3.5, the bound would not grow in a polynomial rate and instead shows a decreasing trend. We conducted experiments and presented results under the same setting as (Nagarajan & Kolter, 2019) in Figure 6. Here we verify two existing generalization bounds from different literature, one from (Bartlett et al., 2017) while another one from (Barron & Klusowski, 2018) in which the former one is composed mainly of product of weight matrices' norm and the latter is comprised of the norm of matrices' product. Empirical results in Figure 6(a) show that under the standard settings the main components of generalization bound in (Bartlett et al., 2017) and Barron & Klusowski (2018) both grows rapidly with respect to the increase in size of the training dataset, as confirmed in (Nagarajan & Kolter, 2019). Another empirical finding in the last column of Figure 6(a) shows that the multiplicative difference between bounds in (Bartlett et al., 2017) and (Barron & Klusowski, 2018) exhibit a constant rate, demonstrating the vacuity of both bounds. However, when measuring the same component under our setting in Figure 6(b), new results showed decreasing bounds as the size of the training dataset increases, concluding the non-vacuity of the associated generalization bounds in our settings.

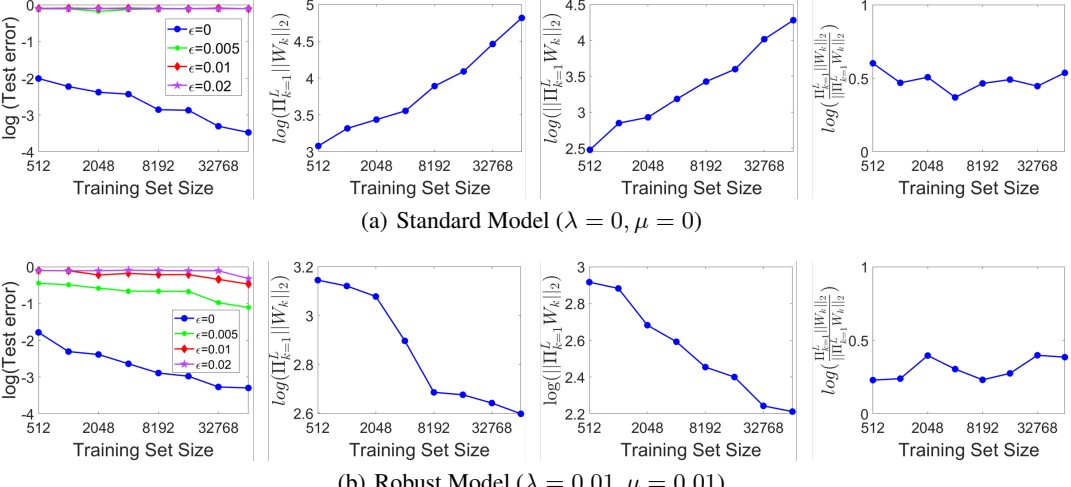

(a) Standard Model ($\lambda = 0, \mu = 0$)

(b) Robust Model ($\lambda = 0.01, \mu = 0.01$)

Figure 6: Statistics associated with generalization bounds for standard model and robust model trained using equation (9). The experiment setting follows Figure 1(a). In the first column, we present test error under weight PGD attack with perturbation level $\epsilon$. The curve with $\epsilon = 0$ (no weight perturbation) corresponds to the standard generalization setting. In the second column, we present the product of spectral norms of the weights matrices, related to bounds in (Bartlett et al., 2017). In the third column, we show spectral norm of the product of the weight matrices, related to bounds in (Barron & Klusowski, 2018). In the fourth column, we show the product of spectral norms of the weights matrices divided by the spectral norm of the product of the weight matrices. Notably, we present each value into the logarithm function. For the standard model, both types of bound increase with respect to training set size and are shown to be vacuous, consistent with the results in (Nagarajan & Kolter, 2019). For the robust model, both types of bound exhibit same decreasing trend as the test error and therefore are non-vacuous. Moreover, these two bounds demonstrate similar scaling behavior (nearly constant log ratio) in both standard and robust models.

As an ablation study, Figure 7 shows the performance of the robust model with $(\lambda = 0.01, \mu = 0)$. That is, training a neural network without the generalization regularization term in equation (9). Unlike the standard model, it is observed that the generalization bounds still show decreasing trend with the test error. This is due to the fact that the robustness term in equation (9), which is induced from our analysis of the worst-case error propagation from weight perturbation, also plays a role in regularizing the network weights, and therefore making the resulting generalization bound non-vacuous.

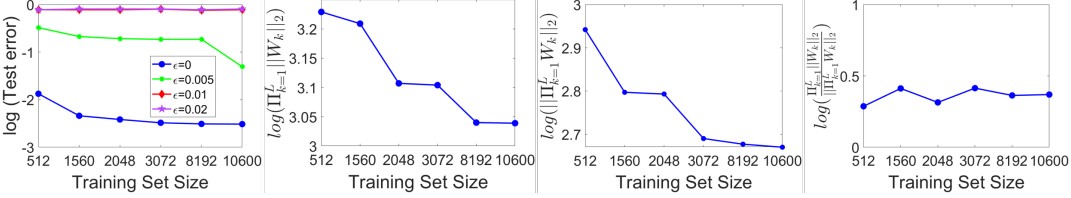

Figure 7: Ablation study of the robust model with $(\lambda = 0.01, \mu = 0)$. The experiment setting is the same as Figure 6. In the absence of the generalization regularization term in equation (9), the generalization bounds still show decreasing trend with the test error.

## F   RUN TIME ANALYSIS

Table 1 reports the per-epoch run time of the models trained with the standard model $(\lambda = 0, \mu = 0)$ and the robust model $(\lambda = 0.01, \mu = 0.1)$ using equation (9). We train both models with 20 epochs and use the same hyperparameters, including setting the SGD optimizer with learning rate as 0.01, and setting the batch size as 32. The cost of training both models are comparable.

|  | Standard model ($\lambda = 0, \mu = 0$) | Robust model ($\lambda = 0.01, \mu = 0.01$) |
|---|---|---|
| Avg. per-epoch run time | 5.125 sec. | 6.69 sec. |

Table 1: Per-epoch run time (in seconds) averaged over 20 epochs with the same hyperparameters for the models trained with $(\lambda = 0, \mu = 0)$ and $(\lambda = 0.01, \mu = 0.1)$ based on equation (9).

