# OpenReview forum: "Formalizing Generalization and Robustness of Neural Networks to Weight Perturbations"
_ICLR.cc/2021/Conference — Reject_

### Official Review · AnonReviewer1 · 2020-10-25
**a trivial application of the existing norm product bound**

**Rating:** 3
**Confidence:** 4

**Review:**

The draft proposes to bound the model generalization by controlling the adversarial perturbation of the model weights. If the weights in the neural network are bounded and the activation function is Lipschitz, the change of output of the network, as well as the Rademacher complexity of the hypothesis class, can be easily controlled.

Connecting perturbation with the generalization is not something new no matter in theory or in practice. This is another draft formulating the network perturbation and generalization so that a norm product bound is derived. There are plenty of previous works on this already, e.g., the work by Neyshabur et. al. (perturb the parameters), and Bartlett et. al. (norm product bound).

The generalization bound proposed in this work is a trivial application of Bartlett’s norm product bound. Perturbing the weights and directly applying norm product leads to a bound not as tight, to some extent, it is mostly vacuous.

Frankly, the method gives a pessimistic norm product bound and ignores all the effects caused by the “alignment” between the internal coefficient matrix and the input vectors, which is crucial in terms of understanding how input signals are handled throughout the network. I would encourage the authors to read some recent work by Barron et.al. which reduces the generalization bound from norm product to product norm.

---

> ### Author Response · Authors · 2020-11-11
> **Asking for reference for "Barron et.al."**
>
> Dear AnonReviewer1,
>
> We sincerely thank you for your review comments. While we are working on revising our submission and preparing for our response, we would like to ask for an explicit reference or link to the work by "Barron et.al." as mentioned in your comments. Thanks!

---

> > ### Comment · AnonReviewer1 · 2020-11-11
> > **the references**
> >
> > Sorry I should make this clear in my comment.
> >
> > The reference link for "Barron et.al." is here:
> > Andrew R. Barron, Jason M. Klusowski, Approximation and Estimation for High-Dimensional Deep Learning Networks,
> > https://arxiv.org/pdf/1809.03090.pdf
> >
> > Another related work along the same line:
> > Ryan Theisen, Jason M. Klusowski, Huan Wang, Nitish S. Keskar, Caiming Xiong, and Richard Socher, Global Capacity Measures for Deep ReLU Networks via Path Sampling, https://arxiv.org/pdf/1910.10245.pdf

---

> ### Author Response · Authors · 2020-11-21
> **Responses to Reviewer 1**
>
> We thank the reviewer for the valuable response and feedback. Here we will address some specific concerns and comments noted by the reviewer.
>
> ---
>
> __*Study of Generalization via the framework of adversarial perturbation*__
>
> We would like to reiterate that unlike papers that study generalization under the standard settings,  in this paper we focus on a fundamentally different perspective and topic --  **generalization and robustness under weight perturbation**, please see general response for detailed settings and introduction.
>
> ---
>
> __*Related works concerning generalization bound and tightness*__
>
> In the review, comments referred to works done by (Barron and Klusowski, 2018) which focus on similar generalization problems studied in (Bartlett et al., 2017). In (Barron and Klusowski, 2018), trenchant observations concerning neural networks with ReLU as activation function were made which later proved to be valuable in deriving a risk upper bound given a certain class of neural networks. Further studies such as (Theisen et al., 2019) extended the method and scenario to a more general case including multiclass classification problem. We here focus on (Barron and Klusowski, 2018) and summarize the main concepts in the work.
>
> Broadly speaking, Barron and Klusowski (2018) first utilized the technique of doubling layer nodes and the homogeneity of ReLU function to aid following analysis of neural networks by describing each neural network in terms of a path distribution. By probabilistically sampling paths, we could construct a representer set which is an empirical path distribution and that by expectation could resemble the original neural network under a certain error threshold. The sampling results could then offer a cover number which in turn implies the bound of generalization.
>
> First of all, we agree that bounds in (Barron and Klusowski, 2018)  can be tighter in **standard generalization** (*but again, this is not our studied setting*) than bounds in (Bartlett et al., 2017) since path sampling offers the sense of global variation which both captures the size of weights within layers and the interactions between successive layers.
>
> However, when applying techniques in (Barron and Klusowski, 2018) under the scenario of generalization under weight perturbation, more subtleties and questions should be considered and studied. For instance, how the global variation (Section 3 in Barron and Klusowski, 2018)  would alter in value when every weight is subjected to perturbation. Once we could analyze the global variation of the network under perturbation, it is then feasible for us to compare the generalization bound (Section 12.2 in Barron and Klusowski, 2018) with the standard generalization setting and derive a solution toward mitigating or narrowing the generalization gap under weight perturbation. Furthermore, by utilizing the scaling invariance nature and optimization analysis (Section 9 in Barron and Klusowski, 2018), one could study the structural shift of parameters when weight perturbations are introduced; namely, what would the optimal subnetwork bound behave when perturbations are present. Since there are aforementioned non-trivial obstacles in directly adapting this technique into our studied problem, and more importantly, the adaptation of the technique is tangential to the research scope and contributions of this paper, we will cite this technical framework but leave the integration as one of our future directions.
>
> That being said, despite there are merits in applying the techniques in (Barron and Klusowski, 2018) to our analysis, we would like to argue that the outcome should not undermine our original contributions in studying and characterizing the generalization behavior of neural networks under weight perturbation. Admittedly, our bound can be tighter if the advanced bounds such as (Barron and Klusowski, 2018) can be integrated, but the results will not fundamentally change our conclusion and insights, as long as the generalization bound is non-vacuous, which was now verified (see **General Response**). The bounds in (Bartlett et al., 2017) and (Barron and Klusowski, 2018) in fact behave similarity (see **Appendix E**) and thus will not affect our conclusion.  We believe the related generalization bounding techniques such as (Barron and Klusowski, 2018) are certainly worthy of further pursuit, but this direction is beyond the scope of this paper. Given that we are considering the setting of generalization under weight perturbation instead of the standard generalization setting, *we believe the evaluation of our work should be independently made and not penalized by not using these results.*

---

> > ### Comment · AnonReviewer1 · 2020-11-23
> > **I will keep my rating but lower my confidence to 4 after seeing the rebuttal**
> >
> > I appreciate all the efforts made by the authors to add experimentation, explanation, and comparison.
> >
> > Still after seeing the rebuttal I would like to keep my rating unchanged. I will lower my confidence to 4 to reflect the empirical observations the authors have made.
> >
> > The authors listed some empirical observations suggesting the bound does not go vacuous with their proposed objective. To me this is not a surprise since in the objective (9) the norms of the coefficient matrices are added explicitly as the regularization. As long as one tweaks the parameter \mu large enough the decreasing bound in section E is expected.
> >
> > On the theory side, the bound is kind of messy and not quite informative.
> >
> > Technique-wise I do not see much novelty either. As I said it is a straightforward application of Bartlett’s bound in the perturbation scenario.

---

> > > ### Author Response · Authors · 2020-11-23
> > > **We respectfully disagree with the comments related to novelty; new experiments are added.**
> > >
> > > We thank Reviewer#1 for the prompt response, and for acknowledging our efforts in adding experimentation, explanation, and comparison.
> > >
> > > However, we are surprised to learn that you feel there is not much technical novelty and our result is a straightforward application of Bartlett’s bound in the perturbation scenario. We respectfully disagree with your comment. In what follows, we reiterate the novelty of our work from three different aspects.
> > >
> > > 1. Only Theorem 4 uses Bartlett’s bound - We would like to emphasize that except for the result in Theorem 4, all other theorems and lemmas do not depend on Bartlett’s bound (which only studies the standard generalization setting). For example, our analysis on worst-case error propagation induced from weight perturbation is completely new and tight. Moreover, we believe science is a continued progress, in which new results were built upon available tools and results. In our case, we used Bartlett’s bound to characterize the generalization gap under weight perturbation based on our novel analysis (e.g. Lemma 2 and Theorem 2). Without our analysis, one cannot apply Bartlett’s bound to study generalization under perturbation. **Comparing our setting versus standard generalization setting, “straightforward application” would imply the minimax formulation (generalization under worst-case weight perturbation) is a straightforward application of the minimization formulation (standard generalization without considering weight perturbation).** Robust machine learning is an active and challenging research topic, and its difference to standard machine learning is well recognized. We hope the reviewer would agree with the essential difference of these two formulations and recognize our contributions.
> > >
> > > 2. The studied generalization under weight perturbation problem setting is novel and practical - In terms of theoretical contributions, to the best of knowledge, we are not aware of any work that provides a theoretical characterization of generalization behavior under weight perturbation. In terms of practical insights, we also proposed an effective training loss function based on our analysis to reduce the generalization gap and empirically verified its effectiveness.
> > >
> > > 3. The empirical results and insights are novel - as all reviewers were curious to know, the generalization bound could be vacuous in our studied setting. However, in the new experiments we showed in Appendix E, we showed that the generalization bound is in fact non-vacuous, avoiding the pessimistic use of the generalization bound. We also want to point out that the robustness loss term from Theorem 3 of equation (9) comes naturally through the consideration of worst-case weight perturbation, which also plays a role in regularizing the weight and in turn makes the generalization bound non-vacuous.  **Following up your comment, we added new experiments in Figure 6 of Appendix E by setting $\mu=0$ and show that the robustness term alone will also make the resulting generalization bounds non-vacuous, which again substantiates the difference between our setting from the standard generalization setting.** Such new insights and effectiveness in reducing the generalization gap under weight perturbation are novel and valuable.
> > >
> > > Ｗe hope this discussion and new experiments can address your concerns.

---

### Official Review · AnonReviewer3 · 2020-10-27
**Interesting theoretical analysis of robustness against weight perturbations**

**Rating:** 7
**Confidence:** 4

**Review:**



In this work, the authors theoretically analyze the robustness against weight perturbations in neural networks. Upper bounds of the pairwise class margin for single-layer, all-layer, and selected-layer perturbation are established. Based on the analysis, the authors propose novel robust surrogate loss functions for 0-1 loss and cross-entropy.  Furthermore,  the authors analyze the Rademacher complexity of the perturbated network with the proposed loss, which leads to generalization bounds based on (Mohri et al. (2018)) and (Bartlett et al. (2017)).

Pros
1.  I think the theoretical analysis part is clear and systematic. Efforts of each term in the bounds are well explained.

2. The analysis is useful for a better understanding of the robustness of networks against weight perturbation.

3. The proposed loss is also interesting.  The explicit upper bound reduces the computation of the maximization step in adversary training of weight perturbations.


Cons
1. The product form in the bounds may grow fast and become loose.  I am not sure about the tightness since quite a few relaxations that rely on triangle inequality and maximum are used during the derivation.  It is better to report the value of each term in the bound in Theorem 2 or 3  for a better understanding of the bound.  The authors can employ the same experiment's setup in Figure 1.

2. Similarly,  I concern about the worst-case error term in the proposed loss in Lemma 2.  If the value of this term is large, the margin term minus the worst-case error will always be negative. Thus, the loss will remain a constant one, which is very harmful to training and optimization.

3. In the experiments, only a simple MNIST dataset is evaluated.  I concern about the performance of the proposed loss in more practical cases. As stated above,  I think the proposed loss may be challenging for optimization.  This phenomenon may become more significant in difficult datasets.

4. In Theorem 1, the definition of z^{N-1} is not given.  It is better to state it clearly to be self-contained.  In Theorem 2, the definition of W* is not given. The definition should be included in the main paper instead of in the appendix.

---

> ### Author Response · Authors · 2020-11-21
> **Responses to Reviewer 3**
>
> We are glad that the reviewer is interested in the study of adversarial robustness and generalization property under weight perturbation and we truly appreciate the careful reading and inspection on our theoretical analysis. We will incorporate reviewer’s feedback to improve our mathematical presentation of main theorems. Here, we would like to clarify and respond to some of the concerns raised by the reviewer.
>
> ---
>
> __*Tightness of the worst-case error bound*__
>
> In terms of the tightness of the error bound caused by weight perturbation, we note that there exist possible scenarios for the worst case error to occur, causing the neural network to misjudge and err by a great extent.
>
> Specifically, using the example in Section 3.2 in the paper, as we trace down the associated inequality bound in equation (5), we see that the first inequality can be achieved when the final weight layer possesses all positive weights and that the row associated with label $\alpha$ is greater than label $\beta$ in all individual entries. Furthermore, as long as we assume that the second weight matrix has equal $\ell_{1}$ norm throughout all rows, we can then tighten the bound to give the worst-case error in equation (6). Similar reasoning can be applied on the multilayer case to offer the worst-case scenario. We have added this discussion in Section 3.2 of our revised version.
>
> Notwithstanding error increases while propagating through layers, we urge still to minimize the worst-case error in order for the model to learn a comprehensive strategy towards weight perturbation.
>
> ---
>
> __*Concerns on surrogate loss function*__
>
> We acknowledge that the reviewer has the correct intuition concerning surrogate loss function. In fact, the surrogate loss function derived in Lemma 2 shows that since error caused by perturbation would be surging rapidly through layers, only small perturbation can be applied in training and practice, permitting the worst-case error term in Lemma 2 to be smaller than the margin term.
>
> *The surrogate loss also implies the difficulty of training robust and generalizable models against large weight perturbations.*
>
>  Other reasoning can be found in the experiment section (Section 4) where Figure 1(b) in the paper demonstrates that even with perturbation as small as $4*10^{-3}$, the performance of the standard model is heavily vandalized, therefore making the model vulnerable under weight perturbation. While the worst-case error term grows with respect to layers and depth, one solution would be to propose it as a regularizer in the loss function, just as reasoning in Section 3.5 shows.
>
> Instead of considering the accuracy and robustness term equally, we take a weighted approach towards the loss function, encouraging the model to continue minimizing the maximum error induced by weight perturbation, while the regularization coefficient $\lambda$ would aid the model to still maintain accuracy in the training process.
>
> On top of that, we thank the reviewer's insightful comments and will include these comments into the paper as an interpretation and discussion of Lemma 2. We have added this discussion in the revised version.
>
> ---
>
> __*Performance on Complex Dataset*__
>
> Although our main contributions are to provide theoretical characterization of the generalization behavior under weight perturbation and our results are not tied to specific datasets, we agree that observing the conclusion also holds on other datasets is meaningful. To be inclusive and reproducible, we plan to release our codes with a generic data loader function, so researchers can test the generalization performance under weight perturbation in future studies. If the reviewer has any particular complex dataset in mind, please let us know and we are happy to run the analysis.
>
> ---
>
> __*Ambiguity of mathematical notations*__
>
> We apologize for that the definition of $z^{N-1}$ in Theorem 1 and definition of $W^*$  Theorem 2 are not fully self-contained, causing ambiguity concerns. Although we’ve first defined the notation of $z^{N-1}$ in Section 2, we understand the comments of the reviewer and have made revisions in the paper to improve the presentation of our main results.

---

> > ### Author Response · Authors · 2020-11-24
> > **Looking forward to Reviewer#3's response**
> >
> > Dear Reviewer#3,
> >
> > As the discussion phase will be ending today, we would like to know if our responses have addressed your initial questions and comments. More importantly, is there anything that you would like to discuss or suggest based on the current version? We look forward to hearing from you!

---

> > > ### Comment · AnonReviewer3 · 2020-11-25
> > > **Response to authors**
> > >
> > >
> > > Sorry for the delayed reply, and thanks for the author's feedback.
> > >
> > > Most of my concerns have been addressed.
> > >
> > > I have some suggestions that may improve the work.
> > >
> > > 1.    I understand the main contributions focus on the theoretical characterization of the generalization behavior under weight perturbation.  However, as pointed out by other reviewers, the theoretical contribution may not be that exciting.
> > >  Thus, I am happy to see some analysis on complex datasets (as shown in my initial Cons. 3), e.g., Tiny-Imagenet,  to demonstrate the practical effectiveness of the proposed loss. This may be helpful to convince other reviewers as the work may have practical influence beyond the theoretical part.
> > > Considering the time limitation, CIFAR100 dataset is also applicable.
> > >
> > >
> > > Minor:
> > > 2. Another suggestion is regarding the proposed loss.
> > >
> > > The following regulation term
> > > $\sum_{m=1}^{L}{ \left( {\log|| ({\bf{W}}^m)^\top ||_{1,\infty}} +{ \log|| {\bf{W}}^m || _{1,\infty} } \right )} $
> > > may be more consistent with the theoretical analysis because the derived bound is in a product form.   This  regulation  term may leads to different optimization landscape.

---

> > > > ### Author Response · Authors · 2020-11-25
> > > > **Thank you for your feedback!**
> > > >
> > > > We appreciate the reviewer's feedback and suggestions. We certainly agree that showing the proposed loss function also works on more complex datasets such as CIFAR100 or Tiny-Imagenet can add more contributions. However, given that the discussion phase is about to close within a couple of hours, it's unlikely that we can provide further updates on this matter due to the constraints on computation time and training over large models. We also like the idea of using the regularization term mentioned by the reviewer. We will further investigate their effects.

---

> > > > ### Author Response · Authors · 2020-11-25
> > > > **Updates and new experiment results in Appendix D.4**
> > > >
> > > > Dear Reviewer#3,
> > > >
> > > > We managed to conduct additional experiments based on your suggestion of training robust models using the alternative regularization term. The results are reported and discussed in Appendix D.4 of the updated version. Comparing to Figure 1(b), the robustness performance using the alternative regularizer is comparable (sometimes can be slightly better) with proper selection of the regularization coefficients, which further verifies the practical utility of leveraging our theoretical insights to reduce generalization gap under weight perturbation, as mentioned by the reviewer.

---

### Official Review · AnonReviewer2 · 2020-10-28
**Robustness Against Weight Perturbations in Neural Networks**

**Rating:** 7
**Confidence:** 4

**Review:**

The paper investigates the effects of weight perturbations on the output margin for multiclass classifcation problems. The paper shows that robustness to weight perturbations can be bounded using the (1,\infty)-norm of the weight matrices. The paper then suggests that a low (1,\infty)-norm of the weight matrices leads to better generalization. Moreover, the robustness against weight perturbation implied by low (1,\infty)-norms of weight matrices should increase the robustness against adversarial perturbations. To support these claims, the paper presents a generalization bound using the (1,\infty)-norm of weight matrices as well as an empirical evaluation using a novel surrogate loss function that, when used in training, is empirically shown to reduce the generalization gap and increase the robustness against adversarial perturbations.

The paper is very well written and the theoretical analysis is sound. My only concern is that the generalization bound derived in theorem 4 might not be very informative or conclusive. The reason is that the Rademacher complexity becomes uninformative in the interpolation regime (i.e., neural network architectures that are complex enough to fit any practical training set perfectly). Since the generalization bound presented here relies on an upper bound on the Rademacher complexity using the already not too tight upper bound of Bartlett, 2017, it could be vacuous. In earlier works on input robustness [1], obtaining meaningful bounds required some covering of the input space with examples rather than a uniform bound over the model space. Since uniform bounds, such as uniform convergence, might not be meaningful at all in the interpolation regime [2] (and thus for most of deep learning), this could imply that the results in Thm. 4 in this paper are not conclusive. In [3], the robustness to weight perturbation was also used as a building block for a potentially non-vacuous generalization bound, so it might be worthwhile to discuss the relation to that paper.

Another line of work that might be worthwhile to discuss is weight noise injection during the training process, which leads to better generalization (e.g., [4]). That is, I would imagine the surrogate loss function is a more effective tool to improve generalization and robustness than random noise injection.

Lastly, it should be discussed how practically applicable the surrogate loss function is. For that, at least a runtime analysis should be provided.

In summary, the paper presents interesting theoretical findings and a potentially practically useful surrogate loss function. The paper is lacking some discussion, but overall I would argue that the paper makes a valid contribution. Thus, I tend to vote for acceptance.

[1] Xu and Mannor. Robustness and generalization. Machine Learning, 2012.
[2] Nagarajan, et al. Uniform convergence may be unable to explain generalization in deep learning. NeurIPS, 2019.
[3] Petzka, et al. Relative Flatness and Generalization in the Interpolation Regime. arxiv preprint, 2020.
[4] An. The effects of adding noise during backpropagation training on generalization performance. Neural computation, 1996.


‐------------ After Discussion ---------
I am still of the opinion that the manuscript is not without flaws: the theoretical result is quite messy, the empirical evaluation is still limited, and the generalization bound could still be vacuous.

However, the authors provided new experiments that indicate that the bound might not be vacuous in the considered setting, though. The authors also provide a preliminary runtime analysis that suggests the costs for using the surrogate loss do not explode (please include a proper runtime analysis in any future version of this paper). Since the authors furthermore did address my main points in my review during the discussion, I have increased my score to 7.

---

> ### Author Response · Authors · 2020-11-21
> **Responses to Reviewer 2**
>
> We thank the reviewer for the valuable feedback and endorsement. We are sincerely glad that the reviewer had a thorough reading and clear understanding of our work. Here we will address some concerns and relevant works raised by the reviewer.
>
> ---
>
> __*Vacuity on Generalization Bound and Relevant Works*__
>
> Following your comment, we have conducted additional experiments (see **Appendix E**) to verify the bound in Bartlett is not vacuous under our setting. We will refer to **General Response** on the issue of vacuity concerning generalization bound for further discussion and empirical results.
>
> Thank you for pointing us to the work of (Petzka et al., 2020).  It offers a fairly intriguing viewpoint into studying generalization under the standard setting. Specifically, Petzka et al. (2020) proposed to separate the neural network into two different components which are feature representation and predictor function respectively in order to fully study generalization behavior of neural networks. This coheres with the traditional machine learning setting where EDA (Exploratory Data Analysis) should be conducted before the actual prediction takes place. Moreover, Petzka et al. (2020) proposes to connect the representativeness (predictor function) measure with implicit feature robustness (feature representation) to give a meaningful generalization bound of modern deep learning models. Overall, we find this work explorable with the integration of our weight perturbation analysis. In vague terms, it is worthwhile to investigate the relation between weight perturbation and the former functions (predictor and feature functions respectively). We believe that the above research topics could prove valuable in providing a more detailed sense of generalization when dealing with models against weight perturbation. We have cited this work in our revised version and will include this as one of the scope for future direction on this specific issue.
>
> ---
>
> __*On the relation between noisy training and surrogate training*__
>
> We appreciate reviewer’s notes on other works (An, 1996) and correct intuition for difference between surrogate training and training with random noise injection. In fact, we found (An, 1996) interesting since the conclusion in (An, 1996) depicting that smaller magnitude with weight would better increase the generalization performance echoes with the theoretical results established in our paper (see **Section 3.4.2** in revised paper). Nevertheless, we note that models trained on worst-case error bound generalizes to a broader scenario, including the event of random noise training with bounded magnitude. We have included this discussion in our revised paper.
>
> ---
>
> __*Run-Time Analysis*__
>
> As a follow-up experiment to  your comment, we show the run-time analysis with an average of one epoch time of the models trained with ($\lambda$ = 0, $\mu$ = 0) and ($\lambda=0.01$, $\mu$ = 0.01). For all experiments, we train models with 20 epochs, use SGD optimizer with learning rate as 0.01, and set batch size as 32. The cost of training a model with our loss is comparable to standard training and is not expensive.
>
> || ($\lambda=0, \mu=0$) | ($\lambda=0.01, \mu=0.1$) |
> | -----------------  |:------------------------------:|:-------------------------:|
> | Avg. per epoch  |      5.125 sec.              |         6.69 sec.         |

---

### Official Review · AnonReviewer4 · 2020-10-28
**A good paper but needs work comparing to prior literature and more experiments to support theory results**

**Rating:** 6
**Confidence:** 3

**Review:**

Summary: The paper discusses learning neural network models under weight parameter perturbations. In particular the paper motivates the use of a new loss function (equation 9) based on the analysis of neural network robustness (Section 3.2, 3.3) and generalization properties (Section 3.4) to perturbations of the weight parameters. Section 4 has experiments supporting the theory that the loss function in equation 9 is robust and has good generalization properties to weight perturbations.

Review: I think the loss function in equation 9 is well motivated and clearly explained. Also the experimental results can be reproduced as the code has been included as part of the submission. Overall the paper is well-written.

But I do have a few concerns regarding the comparison and interpretation of the results with prior work on generalization properties of neural networks. Firstly, the work does not cite some relevant papers to the topic a couple of which I list below:

a.	V. Nagarajan and J.Z. Kolter. Uniform convergence may be unable to explain generalization in deep learning. In NeurIPS 2019.
b.	N. Golowich, A. Rakhlin, and 0. Shamir. Size-independent sample complexity of neural networks. In COLT 2018.

I believe the discussion in Section 2 of Nagarajan and Kolter, 2019 is very relevant to the results in the paper. Nagarajan and Kolter, 2019 show that the norm bounds of the weight matrices increases with the number of samples and hence conclude that the generalization bound results in Bartlett et. al., 2017 are vacuous. The current paper uses the results in Bartlett et. al., 2017 to derive generalization error bounds and additionally has another term dependent on the norms of the weight matrices due to perturbation. Given the discussion in Section 2 of Nagarajan and Kolter, 2018 I am concerned if the bounds obtained in Section 3.2-3.5 are vacuous. I will like to see a discussion and experimental results in light of the observations in Nagarajan and Kolter, 2019.

In the experimental section, I am interested in also knowing the results for the case $\epsilon = 0$ in Figure 1(a). Also, if possible can the authors include a discussion or guidance on the sensitivity of the results to the hyper parameters $\mu$ and $\lambda$? For example the performance seems to get worse when moving from $\lambda = 0.01, 0.125$ to $\lambda = 0.015$ in Figure 1(b).

---

> ### Author Response · Authors · 2020-11-21
> **Responses to Reviewer 4**
>
> Thank you for the valuable feedback. We sincerely express gratitude toward seeing your interests in robustness of neural networks against weight perturbation, and would like to clarify and resolve some concerns in the review.
>
> ---
>
> __*On Vacuous Bound in Bartlett et al. and Related Works*__
>
> We appreciate the reviewer for noting the missing citations of relevant works and had these citations and modifications included in the revised paper and  thank the reviewer for raising concerns on the vacuity of the generalization bound. Following your comment, we have conducted additional experiments (see **Appendix E**) to verify the bound in Bartlett is not vacuous under our setting. We will refer this issue to General Response for detailed discussion and empirical analysis.
>
> For your reference, we also show the new discussion in the revised paper which is included below.
>
> >In (Nagarajan & Kolter, 2019), empirical observations were made to point out the fact that when given increase in the size of the training data, the error bound proposed in (Bartlett et al., 2017) grows rapidly, loosing the ability to describe generalization gap and thus becomes vacuous.  How-ever, we note that under our settings with models trained using the loss function in Section 3.5, the bound would not grow in a polynomial rate and instead shows a decreasing trend.  We conducted experiments and presented results under the same setting as (Nagarajan & Kolter, 2019) in Figure 5.Here we verify two existing generalization bounds from different literature, one from (Bartlett et al.,2017) while another one from (Barron & Klusowski, 2018) in which the former one is composed mainly of product of weight matrices’ norm and the latter is comprised of the norm of matrices’ product. Empirical results in Figure 5(a) show that under the standard settings the main components of generalization bound in (Bartlett et al., 2017) and Barron & Klusowski (2018) both grows rapidly with respect to the increase in size of the training dataset,  as confirmed in (Nagarajan & Kolter,2019). Another empirical finding in the last column of Figure 5(a) shows that the multiplicative difference between bounds in (Bartlett et al., 2017) and (Barron & Klusowski, 2018) exhibit a constant rate,  demonstrating the vacuity of both bounds.  However,  when measuring the same component under our setting in Figure 5(b), new results showed decreasing bounds as the size of the training dataset increases, concluding the non-vacuity of the associated generalization bounds in our settings.
>
> ---
>
> __*Additional Experiments*__
>
> As your comment has suggested, we have conducted additional experiments to show more different settings of Figure 1(a) and a trade-off between $\lambda$ and $\mu$.
>
> To supplement Figure 1(a), we plot the generalization gaps by model trained with $\epsilon$ = 0 (see **Appendix D.1, Figure 2**) when varying the matrix norm regularization coefficient $\mu$. The generalization gaps are obviously lower than others when $\epsilon$ = 0, but their trends are similar.
>
> In order to show the trade-off between coefficients $\lambda$ and $\mu$ (see **Appendix D.3, Figure 4**), we present accuracies under weight PGD attack using perturbation radius of  ($\epsilon$ = 0.01, 0.02) with combinations of $\lambda$ (from 0.01 to 0.015) and $\mu$ (from 0 to 0.05). We find that there is indeed a sweet spot with proper values of $\lambda$ and $\mu$ leading to significantly better robust accuracy.
> When both $\lambda$ and $\mu$ are too large or too small, the robustness of the model will decrease.

---

> > ### Comment · AnonReviewer4 · 2020-11-24
> > **Response to rebuttal**
> >
> > Sorry for the delayed response and thank you to the authors for the effort in addressing all concerns. In light of the observations in Figure 5 Appendix E I am increasing my score to 6. I am still not very excited by the results because I do share concerns raised by Reviewer 1 and also think currently there is not a good discussion around model behaviour in Figure 5. But hopefully the results in the paper can motivate more discussions around the topic.

---

> > > ### Author Response · Authors · 2020-11-24
> > > **Thank you for your feedback!**
> > >
> > > We appreciate the reviewer's feedback and endorsement. Like the reviewer, we also find the results in Figure 5 quite intriguing, which would not happen without the great inputs and suggestions from all reviewers. As the reviewer pointed out, our study on generalization under perturbation motivates new discussions and provides novel observations. We will certainly continue to explore this direction to better understand the model behavior.

---

### Author Response · Authors · 2020-11-13
**Clarification on “generalization under weight perturbation” and Discussion on plans to verify non-vacuous generalization bound under weight perturbation**

We thank all reviewers for their constructive comments. While we are preparing the revised version of our submission and the suggested experiments, we would like to leverage the interactive discussion function of OpenReview for efficient communication, concept clarification, and feedback solicitation for our planned experiments for verifying our generalization bound under weight perturbation is non-vacuous.

$\bullet$ $\textbf{Fundamental difference between standard generalization setting and ours:}$

While reviewers have correctly and clearly summarized our studied generalization setting, we would like to use this opportunity to reiterate the fundamental difference of our setting from current studies, in order to facilitate the following discussion. Our goal is to characterize the generalization behavior of a neural network *when its weights are subject to perturbation*, which is different from the standard setting studying generalization that does not consider generalization under weight perturbation. Notably, as we discussed in the related works section, Keskar et al. (2017) proposed the sharpness as a generalization metric using weight sensitivity, but the objective is to study *standard* generalization, instead of “generalization under weight perturbation” (our setting). That being said, we fully agree that the mentioned works in the reviews are related and we can use their experiment design to further verify our proposed framework, which will be incorporated and discussed in the revised version.

$\bullet$  $\textbf{Our plans to verify non-vacuous generalization bound under weight perturbation:}$

All reviewers shared a common concern about our generalization bound could be vacuous, based on the argument that part of our theoretical framework is built upon the norm-product bound in (Bartlett et al. 2017), and especially the recent results of vacuous bounds (Figure 1 and Section 2) reported in the paper “Uniform convergence may be unable to explain generalization in deep learning” by Nagarajan and Kolter, in the setting of *standard generalization*. The results show that for standard generalization, as the number of training samples increases, the norm product also increases at a similar rate while the actual test error decreases, suggesting that the norm product associated with standard generalization is vacuous.

We fully understand the reviewers’ concern and plan to conduct similar experiments as in (Nagarajan and Kolter, 2019) to verify whether our generalization bound is vacuous or not *under weight perturbation*. We believe our generalization bound under weight perturbation will not be vacuous due to the following reasons:

1) Since we study the generalization behavior under weight perturbation (i.e., $\epsilon \neq 0$), a robust model should be less sensitive to weight perturbation. Unlike the standard generalization setting (i.e. $\epsilon = 0$), confining weight sensitivity is necessary in our setting to improve generalization under weight perturbation.  More importantly, as shown in Figure 1 (b) of our submission, even though models trained using standard loss ($\lambda=\mu=0$) can be sensitive to weight perturbation and have large generalization error under weight perturbation, our results show that models trained using our proposed loss in equation (9)  (all curves other than the black curve) can reduce the generalization gap *under weight perturbation*, suggesting the practical utility of our analysis.

2) Unlike the standard generalization setting in which the norm of the network weights are not confined, our theory-driven loss as shown in equation (9) actually penalizes the weight norm. Therefore, the weight norm product trained using our proposed loss is expected to exhibit a sublinear rate in the number of training samples, making our generalization bound non-vacuous. Our intuition also echoes the statement in (Nagarajan and Kolter, 2019), which we quote: “At the same time, we must add the disclaimer that our results do not preclude the fact that uniform convergence may still work if GD is run with explicit regularization (such as weight decay)” In our setting, we show theoretically and empirically that confining weight norm is necessary to reduce generalization gap under weight perturbation.

To verify our claim of non-vacuous generalization bound under weight perturbation, we plan to use the experiment setup in Figure 2 of (Nagarajan and Kolter, 2019), train models using our theory-driven loss in equation (9), and report the weight norm associated with our derived generalization bound, as well as the actual test error, when we vary the number of training samples. We will post another response when such results are available. In the meantime, we would like to solicit feedback from reviewers in terms of our experiment design, and in particular, any other suggestion to validate our generalization bound is vacuous or not.

---

### Author Response · Authors · 2020-11-21
**General Response on Vacuous Generalization Bound**

We would like to first thank all reviewers for their valuable feedback, as insightful comments and critical discussions both play an indispensable role in the process of research development. We will address specific questions under individual review sections. Here, we will respond to common concerns. We also marked our changes in the revised version in blue color.

### __Discourse on Vacuous Generalization Bound__

As mentioned in most reviews and discussed as a topic in (Nagarajan and Kolter, 2019), the risk bound proposed in (Neyshabur et al., 2018), (Bartlett et al., 2017) offers a false sense of risk since empirical observations in (Nagarajan and Kolter, 2019) suggests that given increase in the size of training dataset, the weight norm of each layer would be surging rapidly thus render the aforementioned bounds vacuous. However, in the setting of (Nagarajan and Kolter, 2019),  standard form of training was taken (without weight decay, dropouts…etc) into consideration, which is __fundamentally different__ from our settings that take weight perturbation into consideration.

While most papers discussing standard generalization propose to bound the gap by means of introducing perturbation to the network. We in turn investigate a fundamentally different topic. In our settings, we start by investigating the performance of a given __neural network under weight perturbation__. Therefore, a natural approach which was later used as a tool for measuring robustness of neural networks is the margin of a neural network given a certain data input and correct label. Moreover, after deriving the bound on worst-case error propagation, we turn to hone in on a slightly deeper topic. Given that the network is trained to handle perturbations applied on weight parameters, will the generalization gap be widen and if so to what extent? Namely, we aim to study whether or not *the generalization ability is compromised once robustness to weight perturbation is taken into account in the training process* , as well as possible solutions to mitigate this effect.

As reviews noted empirical observations in (Nagarajan and Kolter, 2019), we conducted experiments and presented results in the same setting as (Nagarajan and Kolter, 2019) and reported the results in (see __Appendix E, Figure 5__). As the experiments have shown that __under the standard settings__ the main components of generalization bound in (Bartlett et al., 2017) (product of matrix norm) indeed grows with respect to the increase in size of the training dataset. However, when we measure the multiplicative difference between two bounds, one in (Bartlett et al., 2017)  while another in (Barron and Klusowski, 2018), no major difference in their scaling behavior is spotted. Therefore, our results lead to the conclusion of the __vacuity__ of both (Bartlett et al., 2017)  and (Barron and Klusowski, 2018) bounds __in standard generalization setting__. However, when measuring the same component __under our setting__ (trained with the loss function proposed in equation (9)), the new results (see __Appendix E, Figure 5__) showed __decreasing bound__ as the size of the training dataset increases, concluding the non-vacuity in our settings. As an echo to both our theoretical analysis and experiments concerning generalization bounds, we quote a statement from (Nagarajan and Kolter, 2019) which describes that

>“At the same time, we must add the disclaimer that our results do not preclude the fact that uniform convergence may still work if GD is run with explicit regularization (such as weight decay)” .

Thus to this point, we’ve shown theoretically and empirically that confining the weight norm is necessary to reduce the generalization gap under weight perturbation. More importantly, the generalization bound in our setting is shown to be __non-vacuous__, and therefore we believe the reviewers’ common concerns are well addressed.

---

### Author Response · Authors · 2020-11-23
**Looking forward to replies from Reviewer #2, #3, #4**

Dear Reviewer #2, #3, #4,

As Discussion Stage 2 is about to end, we are looking forward to your reply. Please kindly let us know if our response has addressed your initial questions. We appreciate your input and are happy to discuss any follow-up questions. Thank you!

We also thank Reviewer #1 for engaging in the discussion. We hope our reply has addressed your concerns.

---

### Decision · Program_Chairs · 2021-01-07
**Final Decision**

**Decision:**

Reject

**Comment:**

The authors study empirically and theoretically the behavior of neural networks under $l_\infty$-perturbations on the weight matrix.
For this purpose they first derive bounds on the logit-layer of the neural networks under perturbuations of a single or all layers. Then they propose to merge this bound (which depends on the product of the weight matrices) into a margin-based loss function/cross-entropy-loss and suggest to optimize this bound while simultaneously penalizing the 1,infty-norm of the weight matrices (which appears in the bound). Furthermore they derive a generalization bound for the robust error under weight perturbations.

There was a discussion among the reviewers over this paper. While several ones appreciated the setting, there was concern about that the bound is potentially vacuous and that the theoretical results are "messy". One reviewer criticized heavily the bound as not very useful and overly pessimistic.

This paper is in my point of view borderline but I argue for rejection. There are several reasons for this
- the motivation for this paper remains unclear. While the authors argue that a network could be attacked by changing logical values, this seems at the moment unrealistic as if the attacker has already hacked into the system much more harm can be caused in a much easier way e.g. by directly changing the output of the network. But even if one considers adversarial bit error attacks to be realistic, then the threat model would be completely different from $l_\infty$ and would rather be like $l_0$ (with potential unbounded changes if the network is not quantized). If the target is to study that more flat minima generalize better, then the target would not be a bound on the robust error but a bound on the normal test error which integrates an upper bound which measures "flatness" of the function. As the derived Theorem 4 contains a term where one has to take the supremum over the product of matrices over all functions in the derived function class, this is not true for the derived bound.
Thus I don't see why this bound is related to either of these two motivating topics.

- the bound is incomplete in the sense that it contains the sup_f of the product of 1,infty norms of the weight matrices. Why
did the authors not try to upper bound this term over the chosen function class? Even better would clearly to derive a bound on the Rademacher complexity in terms of the norms which are actually appearing in the bound. In the current way the terms in the bound and the chosen function class are mis-aligned.

- From a practical perspective the resulting loss is not useful for training deeper models (in the experiments a four layer network is used) as the product of the norm of the weight matrices grows exponentially in depth. Moreover, as pointed out the IBP bounds of Weng et al (2020) are much tighter than the bounds derived in this paper (and even IBP bounds are loose)

- The experiments suggests that one gets a minor improvement in robustness while having to suffer from a significant drop in test accuracy (Figure 1b). Regarding the achieved robustness for epsilon=0.01 (it remains completely unclear how this noise model relates to the normal size of the model weights) one gets a robust error of around 30% on MNIST. This is much worse than what has been achieved for MNIST with much larger input perturbations (epsilon=0.3)

- The authors are missing an assumption on the activation function. With just non-negativity, monotonicity and 1-Lipschitz
the upper bound in A.2.1 (original version) cannot be derived. I guess you are implicitly assuming that rho(0)=0 but this assumption is not stated. There are other typos in the main text. In fact in the original version Theorem 4 did not contain the term
depending on b_h and s_j - this term was added in the revised version but not highlighted as all other changes.

In total there are too many open issues here. While I appreciate the hard work the authors put into the author rebuttal, I think that this paper needs a major revision before it can be published.